# AsEP: Benchmarking Deep Learning Methods for Antibody-specific Epitope Prediction

**Chu'nan Liu**[*]
Structural Molecular Biology
University College London
United Kindom

**Lilian Denzler**[†]
Structural Molecular Biology
University College London
United Kindom

**Yihong Chen**[†]
Centre for Artificial Intelligence
University College London
United Kindom

**Andrew Martin**[*]
Structural Molecular Biology
University College London
United Kindom

**Brooks Paige**[*]
Centre for Artificial Intelligence
University College London
United Kindom

## Abstract

Epitope identification is vital for antibody design yet challenging due to the inherent variability in antibodies. While many deep learning methods have been developed for general protein binding site prediction tasks, whether they work for epitope prediction remains an understudied research question. The challenge is also heightened by the lack of a consistent evaluation pipeline with sufficient dataset size and epitope diversity. We introduce a filtered antibody-antigen complex structure dataset, *AsEP* (Antibody-specific Epitope Prediction). *AsEP* is the largest of its kind and provides clustered epitope groups, allowing the community to develop and test novel epitope prediction methods and evaluate their generalisability. *AsEP* comes with an easy-to-use interface in Python and pre-built graph representations of each antibody-antigen complex while also supporting customizable embedding methods. Using this new dataset, we benchmark several representative general protein-binding site prediction methods and find that their performances fall short of expectations for epitope prediction. To address this, we propose a novel method, *WALLE*, which leverages both unstructured modeling from protein language models and structural modeling from graph neural networks. *WALLE* demonstrate up to 3-10X performance improvement over the baseline methods. Our empirical findings suggest that epitope prediction benefits from combining sequential features provided by language models with geometrical information from graph representations. This provides a guideline for future epitope prediction method design. In addition, we reformulate the task as bipartite link prediction, allowing convenient model performance attribution and interpretability. We open source our data and code at `https://github.com/biochunan/AsEP-dataset`.

---

[*]Address correspondence to: `chunan.liu@ucl.ac.uk`, `b.paige@ucl.ac.uk`, `andrew.martin@ucl.ac.uk`
[†]Equal contribution

38th Conference on Neural Information Processing Systems (NeurIPS 2024) Track on Datasets and Benchmarks.

# 1 Introduction

Antibodies are specialized proteins produced by our immune system to combat foreign substances called antigens. Their unique ability to bind with high affinity and specificity sets them apart from regular proteins and small-molecule drugs, making them increasingly popular in therapeutic engineering. While the community is shifting towards computational antibody design based on pre-determined epitopes (Jin et al., 2022; Zhou et al., 2024; Bennett et al., 2024), accurate prediction of epitopes themselves remains underexplored. Precise epitope identification is essential for understanding antibody-antigen interactions and antibody functions, as well as streamlining antibody engineering. The task remains challenging due to multiple factors, including the lack of comprehensive datasets, limited interpretability, and low generalizability (Akbar et al., 2022; Hummer et al., 2022). Existing datasets are limited in size, e.g. only 582 complexes in Bepipred-3.0 (Clifford et al., 2022), and often exhibit disproportionate representation among different epitopes. Current methods perform poorly on epitope prediction (Cia et al., 2023), with a ceiling MCC (Matthew's Correlation Coefficient) of 0.06. Besides, recent advances in graph learning algorithms, along with an increase in available antibody structures in the Protein Data Bank (PDB) (Berman et al., 2003), highlight the need to reevaluate current methods and establish a new benchmark dataset for predicting antibody-antigen interactions.

We approach the problem as a bipartite graph link prediction task, where the goal is to directly identify connections between two distinct graphs representing the antibody and antigen. Unlike conventional link prediction tasks (Nickel et al., 2016; Trouillon et al., 2016; Zhang & Chen, 2018; Chen et al., 2021), aiming to predict links within an individual graph (potentially with multiple relation types), such as in a protein-protein interaction network, our approach predicts links across a pair of molecular graphs. Our model, WALLE, is designed to predict fine-grained residue-residue interactions, i.e. bipartite links between the antibody and antigen, while it can also adapt to solve the coarser node classification task, distinguishing binding nodes from non-binding ones. Since most existing methods focus on predicting protein binding sites (i.e. node classification), we first benchmark this binding node classification task for straightforward comparisons with these methods. Moreover, we include WALLE's performance on the bipartite link prediction task as a baseline for future work.

# 2 Previous Work

Accurate epitope prediction for antibody design remains challenging due to the complexity of antibody-antigen interactions and limitations of existing datasets and methods. Several computational approaches have been developed, but they often fall short in terms of accuracy and applicability. Here, we present a representative, yet non-exhaustive, set of state-of-the-art methods.

**EpiPred** (Krawczyk et al., 2014) implements a graph-based antibody-antigen specific scoring function that considers all possible residue-residue pairs at the interface between antibody and antigen structures. It samples surface patches from the antigen structure and selects the highest-ranked patch as the predicted set of epitope residues.

**ESMFold**(Lin et al., 2023) is a protein language model based on ESM2 (Lin et al., 2023), and its folding head was trained on over 325 thousand protein structures. It achieves comparable performance to AlphaFold2 (Jumper et al., 2021) and is included in our benchmark due to its faster processing.

We also include methods that consider only antigens:

**ESMBind** (Schreiber, 2023) is a language model that predicts protein binding sites for single protein sequence inputs. It is fine-tuned based on ESM2 (Lin et al., 2023) using Low-Rank Adaptation (Hu et al., 2021) on a dataset composed of more than 200 thousand protein sequences with annotated binding sites.

**MaSIF-site** (Gainza et al., 2020) is a geometric deep learning method that predicts binding sites on the surface of an input protein structure. It converts antigen surfaces into mesh graphs, with each mesh vertex encoded with geometric and physicochemical descriptors, and predicts binding sites as a set of mesh vertices.

**PECAN and EPMP** (Pittala & Bailey-Kellogg, 2020; Vecchio et al., 2021) are graph neural networks that predict epitope residues taking antibody-antigen structure pairs as input. They use position-specific scoring matrices (PSSM) as node embeddings and graph attention networks to predict the binary labels of nodes in the antigen graph. For details, we refer readers to Appendix A.1.

Two **complementary surveys** are notable:

Zhao et al. (2024) benchmarked docking methods like ZDOCK (Pierce et al., 2011), ClusPro (Kozakov et al., 2017), and HDOCK (Yan et al., 2017), and Alphafold-Multimer (Evans et al., 2022) on a set of 112 antibody-antigen complexes. They showed that all docking methods gave a success rate of $8.0\%$ at most if using the top 5 decoys; AlphaFold-Multimer showed a better performance with a $15.3\%$ success rate, so we included **AlphaFold-Multimer** (version 2.3) but benchmarked on a separate subset of the proposed dataset (AsEP) according to its training data cutoff date.

Cia et al. (2023) focused on epitope prediction using a dataset of 268 complexes, defining epitope residues as having at least a $5\%$ change in relative solvent accessibility upon complex formation. They benchmarked various methods, finding existing methods insufficient for accurate epitope prediction.

# 3 Problem Formulation

Antibody-antigen interaction is important for analyzing protein structures. The problem can be formulated as a bipartite graph link prediction task. The inputs are two disjoint graphs, an antibody graph $G_A = (V_A, E_A)$ and an antigen graph $G_B = (V_B, E_B)$, where $V_x$ is the vertice set for graph $x$ and $E_x$ is the edge set for graph $x$. Since neural networks only take continuous values as input, we encode each vertex into a vector with the function $h : V \to \mathbb{R}^D$. The design choice of the encoding function depends on the methods. For example, $h$ can be a one-hot encoding layer or pretrained embeddings given by a protein language model. We use different encoding functions for antibodies and antigens: $h_A : V_A \to \mathbb{R}^{D_A}$, and $h_B : V_B \to \mathbb{R}^{D_B}$.

In addition, $E_A \in \{0,1\}^{|V_A| \times |V_A|}$ and $E_B \in \{0,1\}^{|V_B| \times |V_B|}$ denote the adjacency matrices for the antibody and antigen graphs, respectively. In this work, the adjacency matrices are calculated based on the distance matrix of the residues. Each entry $e_{ij}$ denotes the proximity between residue $i$ and residue $j$; $e_{ij} = 1$ if the Euclidean distance between any non-hydrogen atoms of residue $i$ and residue $j$ is less than $4.5\text{Å}$, and $e_{ij} = 0$ otherwise (See example in Figure 1. The antibody graph $G_A$ is constructed by combining the CDR residues from the heavy and light chains of the antibody, and the antigen graph $G_B$ is constructed by combining the surface residues of the antigen. The antibody and antigen graphs are disjoint, i.e., $V_A \cap V_B = \emptyset$.

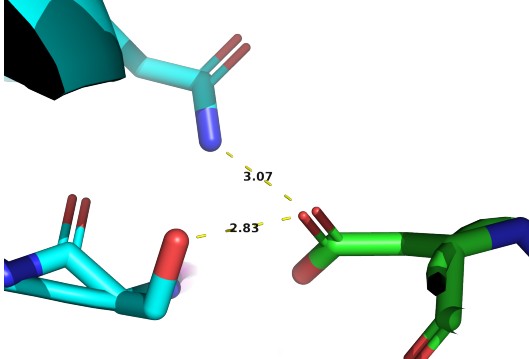

Figure 1: An example illustrating interacting residues. The two dashed lines indicate distances between non-hydrogen atoms from different interacting residues across two protein chains, with each chain's carbon atoms colored cyan and green.

We consider two subtasks based on these inputs.

**Epitope Prediction** Epitopes are the regions on the antigen surface recognized by antibodies; in other words, they are a set of antigen residues in contact with the antibody and are determined from the complex structures using the same distance cutoff of $4.5\text{Å}$ as aforementioned. For a node in the antigen graph $v \in V_B$, if there exists a node in the antibody graph $u \in V_A$ such that the distance between them is less than $4.5\text{Å}$, then $v$ is an epitope node. Epitope nodes and the remaining nodes in $G_B$ are assigned labels of 1 and 0, respectively. The first task is then a node classification within the antigen graph $G_B$ given the antibody graph $G_A$.

This classification takes into account the structure of the antibody graph, $G_A$, mirroring the specificity of antibody-antigen binding interactions. Different antibodies can bind to various antigen locations, corresponding to varying subsets of epitope nodes in $G_B$. This formulation differs from conventional antigen-only epitope prediction that does not consider the antibody structure and ends up predicting the likelihood of the subset of antigen nodes serving as epitopes, such as ScanNet (Tubiana et al., 2022), MaSIF (Gainza et al., 2020). The goal is to develop a binary classifier $f : V_B \rightarrow \{0, 1\}$ that takes both antibody and antigen graphs as input and predicts the labels for antigen nodes:

$$f(v\,;G_B, G_A) = \begin{cases} 1 & \text{if } v \text{ is an epitope;} \\ 0 & \text{otherwise.} \end{cases} \tag{1}$$

**Bipartite Link Prediction** The second task takes it further by predicting concrete interactions between nodes in $G_A$ and $G_B$, resulting in a bipartite graph that represents these antibody-antigen interactions. Moreover, this helps attribute the model performance to specific interactions at the molecular level and provide more interpretability. Accurately predicting these interactions is critical for understanding the binding mechanisms and for guiding antibody engineering. We model the antibody-antigen interaction as a bipartite graph $K_{m,n} = (V_A, V_B, E)$ where $m = |V_A|$ and $n = |V_B|$ denote the numbers of nodes in the two graphs, respectively, and $E$ denotes all possible inter-graph links. In this bipartite graph, a node from the antibody graph is connected to each node in the antigen graph via an edge $e \in E$. The task is then to predict the label of each bipartite edge. If the residues of a pair of nodes are located within $4.5\text{Å}$ of each other, referred to as *in contact*, the edge is labeled as 1; otherwise, 0. For any pair of nodes, denoted as $(v_a, v_b)\,\forall v_a \in V_A, v_b \in V_B$, the binary classifier $g : K_{m,n} \rightarrow \{0, 1\}$ is formulated as below:

$$g(v_a, v_b; K_{m,n}) = \begin{cases} 1 & \text{if } v_a \text{ and } v_b \text{ are in contact} \\ 0 & \text{otherwise.} \end{cases} \tag{2}$$

## 4 AsEP Dataset

We present our dataset AsEP of filtered, cleaned and processed antibody-antigen complex structures. It is the largest collection of antibody-antigen complex structures to our knowledge. Antibodies are composed of two heavy chains and two light chains, each of which contains a variable domain (areas of high sequence variability) composed of a variable heavy (VH) and a variable light (VL) domain responsible for antigen recognition and binding (Chothia & Lesk, 1987). These domains have complementarity-determining regions (CDR, Figure 2 top blue, yellow, and red regions), which are the primary parts of antibodies responsible for antigen recognition and binding.

### 4.1 Antibody-antigen complexes

We sourced our initial dataset from the Antibody Database (AbDb) (Ferdous & Martin, 2018), dated 2022/09/26, which contains $11,767$ antibody files originally collected from the Protein Data Bank (PDB) (Berman et al., 2003). We extracted conventional antibody-antigen complexes that have a VH and a VL domain with a single-chain protein antigen, and there are no unresolved CDR residues due to experimental errors, yielding $4,081$ antibody-antigen complexes. To ensure data balance, we removed identical complexes using an adapted version of the method described in Krawczyk et al. (2014). We clustered the complexes by antibody heavy and light chains followed by antigen sequences using MMseqs2 (Steinegger & Söding, 2017). We retained only one representative complex for each unique cluster, leading to a refined dataset of $1,725$ unique complexes. Two additional complexes were manually removed; CDR residues in the complex 6jmr_1P are unknown (labeled as 'UNK') and it is thus impossible to build graph representations upon this complex; 7sgm_0P was also removed because of non-canonical residues in its CDR loops. The final dataset consists of $1,723$ antibody-antigen complexes. For detailed setup and processing steps, please refer to Appendix A.3.

### 4.2 Convert antibody-antigen complexes to graphs

These $1,723$ files were then converted into graph representations, which are used as input for WALLE. In these graphs, each protein residue is modeled as a vertex. Edges are drawn between pairs of

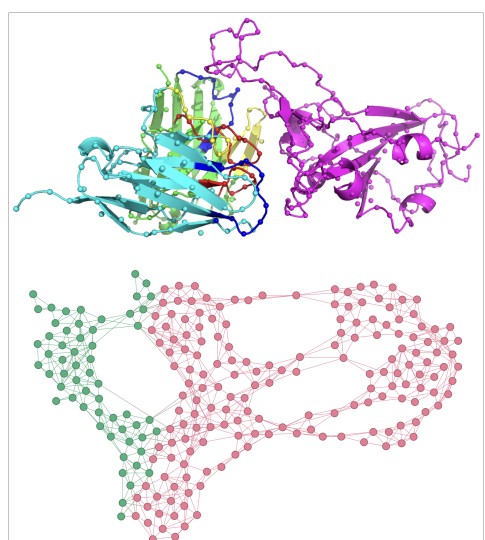

Figure 2: Graph visualization of an antibody-antigen complex. **Top**: the molecular structure of an antibody complexed with the receptor binding domain of SARS-Cov-2 virus (PDB code: 7KFW), the antigen. Spheres indicate the alpha carbon atoms of each amino acid. Color scheme: the antigen is colored in magenta, the framework region of the heavy and light chains is colored in green and cyan and CDR 1-3 loops are colored in blue, yellow, and red, respectively. **Bottom**: the corresponding graph. Green vertices are antibody CDR residues and pink vertices are antigen surface residues.

residues if any of their non-hydrogen atoms are within $4.5$Å of each other, adhering to the same distance criterion used in PECAN (Pittala & Bailey-Kellogg, 2020).

**Exclude buried residues** In order to utilize structural information effectively, we focused on surface residues, as only these can interact with another protein. Consequently, we excluded buried residues, those with a solvent-accessible surface area of zero, from the antigen graphs. The solvent-accessible surface areas were calculated using DSSP (Kabsch & Sander, 1983) via Graphein (Jamasb et al., 2021). It is important to note that the number of interface nodes are much smaller than the number of non-interface nodes in the antigen, making the classification task more challenging.

**Exclude non-CDR residues** We also excluded non-CDR residues from the antibody graph, as these are typically not involved in antigen recognition and binding. This is in line with the approach adopted by PECAN (Pittala & Bailey-Kellogg, 2020) and EPMP (Vecchio et al., 2021). Figure 2 provides a visualization of the processed graphs.

**Node embeddings** To leverage the state-of-the-art protein language models, we generated node embeddings for each residue in the antibody and antigen graphs using AntiBERTy (Ruffolo et al., 2021) (via IgFold (Ruffolo et al., 2023) package) and ESM2 (Lin et al., 2022) (`esm2_t12_35M_UR50D`) models, respectively. In our dataset interface package, we also provide a simple embedding method using one-hot encoding for amino acid residues. Other node embedding methods can be easily incorporated into our dataset interface.

### 4.3 Dataset split

We propose two types of dataset split settings. The first is a random split based on the ratio of epitope to antigen surface residues, $\frac{\#\text{epitope nodes}}{\#\text{antigen nodes}}$; the second is a more challenging setting where we split the dataset by epitope groups. The first setting is straightforward and used by previous methods, while the second setting requires the model to generalize to unseen epitope groups.

**Split by epitope to antigen surface ratio** As aforementioned, the number of non-interface nodes in the antigen graph is much larger than the number of interface nodes. While epitopes usually have a limited number of residues, typically around $14.6 \pm 4.9$ amino acids (Reis et al., 2022), the antigen surface may extend to several hundred or more residues. The complexity of the classification

task, therefore, increases with the antigen surface size. To ensure similar complexity among train, validation, and test sets, we stratified the dataset to include a similar distribution of epitope to non-epitope nodes in each set. Table S3 shows the distribution of epitope-to-antigen surface ratios in each set. This led to 1383 antibody-antigen complexes for the training set and 170 complexes each for the validation and test sets. The list of complexes in each set is provided in the Supplementary Table `SI-split-epitope-ratio.csv`.

**Split by epitope groups** This is motivated by the fact that antibodies are highly diverse in the CDR loops and by changing the CDR sequences it is possible to engineer novel antibodies to bind different sites on the same antigen. This was previously observed in the EpiPred dataset where Krawczyk et al. (2014) tested the specificity of their method on five antibodies associated with three epitopes on the same antigen, *hen egg white lysozyme*.

We inlcude 641 unique antigens and 973 epitope groups in our dataset. We include multi-epitope antigens. For example, there are 64 distinct antibodies that bind to coronavirus spike protein. We can see that different antibodies bind to different locations on the same antigen. Details of all epitope groups are provided in the Supplementary Table `SI-AsEP-entries.csv`. We then split the dataset into train, validation, and test sets such that the epitopes in the test set are not found in either train or validation sets. We used an 80%/10%/10% split for the number of complexes in each set. This resulted in 1383 complexes for the training set and 170 complexes for the validation and test sets. The list of complexes in each set is provided in the Supplementary Table `SI-split-epitope-group.csv`.

**User-friendly Dataset Interface** We implemented a Python package interface for our dataset using PyTorch Geometric (Fey & Lenssen, 2019). Users can load the dataset as a PyTorch Geometric dataset object and use it with PyTorch Geometric's data loaders. We provide an option to load node embeddings derived from AntiBERTy and ESM2 or simply one-hot embeddings. Each data object in the dataset is a pair of antibody and antigen graphs; both node- and edge-level labels are provided, and the node-level labels are used for the epitope prediction task.

# 5    WALLE: A Hybrid Method for Epitope Prediction

Alongside our dataset interface, we also provide a new method named WALLE. It takes as input a pair of antibody and antigen graphs, constructed as detailed above for the AsEP dataset, and makes node-level and edge-level predictions.

**Graph Modules** The architecture of WALLE incorporates graph modules that process the input graphs of antibody and antigen structures, as depicted in Figure 3). Inspired by PECAN and EPMP, our model treats the antibody and antigen graphs separately, with distinct pathways for each. The antibody graph is represented by node embeddings $X_A$ with a shape of $(M, D_A)$ and an adjacency matrix $E_A$, while the antigen graph is described by node embeddings $X_B$ with a shape of $(N, D_B)$ and its corresponding adjacency matrix $E_B$. Both antibody and antigen graph nodes are first projected into the dimensionality of 128 using fully connected layers. The resulting embeddings are then passed through two GNN layers consecutively to refine the features and yield updated node embeddings $X'_A$ and $X'_B$ with a reduced dimensionality of $(M, 64)$. The output from the first GNN layer is passed through a ReLU activation function. Outputs from the second GNN layer are directly fed into the *Decoder* module. These layers operate independently, each with its own parameters, ensuring that the learned representations are specific to the antibody or the antigen. The use of separate graph modules for the antibody and antigen allows for the capture of unique structural and functional characteristics pertinent to each molecule before any interaction analysis. This design choice aligns with the understanding that antibodies and antigens have distinct roles in their interactions and that their molecular features should be processed separately.

**Combining unstructured (sequential) with structural modeling**    The embedding size, $D_A$ and $D_B$, are determined by the pre-trained protein language model (PPLM). We extracted embeddings from the final layer outputs of each PPLM as node features for our GNNs. We experimented with different combinations of graph neural network (GNN) architectures and PPLM embeddings. We assessed commonly used graph modules, Graph Convolutional Network (GCN) (Kipf & Welling, 2017), Graph Attention Network (GAT) (Veličković et al., 2018), and GraphSAGE networks (Hamilton et al., 2018); and PPLMs including, AntiBERTy (Ruffolo et al., 2021), ESM2-35M (`esm2_t12_35M_UR50D`), ESM2-650M (`esm2_t33_650M_UR50D`) (Lin et al., 2022).

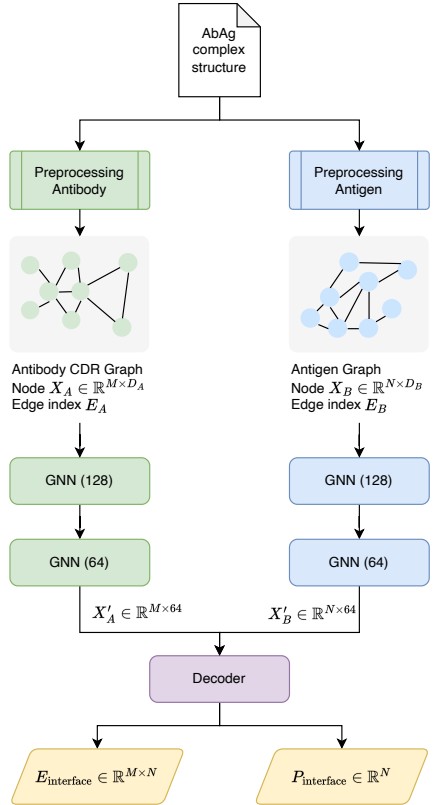

Figure 3: A schematic of the preprocessing step that turns an input antibody-antigen complex structure into a graph pair and the model architecture of WALLE.

**Decoder** We used a simple decoder to predict binary labels for edges between the antibody and antigen graphs. This decoder takes pairs of node embeddings, output by the graph modules, as input and predicts the probability of each edge. During hyperparameter tuning, we compute the final logits by either taking the inner product of the antibody and antigen embeddings or concatenating them and passing them through a single linear layer with dropout. A sigmoid function is applied to produce the final probabilities. An edge is assigned a binary label of $1$ if the predicted probability exceeds $0.5$ or $0$ otherwise. This is shown as the *Decoder* module in Figure 3. For the epitope prediction task, we convert edge-level predictions to node-level by summing the predicted probabilities of all edges connected to an antigen node. We assign the antigen node a label of $1$ if the number of connected edges surpasses a set threshold or $0$ otherwise. This threshold is treated as a hyperparameter and optimized during experimentation.

**Implementation** We used PyTorch Geometric (Fey & Lenssen, 2019) framework to build our model. The graph modules are implemented using *GCNConv*, *GATConv*, *SAGEConv* modules from PyTorch Geometric. We trained the model to minimize a loss function consisting of two parts: a weighted binary cross-entropy loss for the bipartite graph link reconstruction and a regularizer for the number of positive edges in the reconstructed bipartite graph. We used the same set of hyperparameters and loss functions for both dataset settings. The loss function and hyperparameters are described in detail in Appendix A.6. We report only the best performance of different combinations of graph modules and pre-trained language model embeddings in Table 1 (refer to Appendix B for the performance of all combinations.

## 6    Results and Discussion

For epitope residue prediction, we evaluated each method on both dataset split settings using the metrics described in Appendix A.2. The results are summarized in Table 1a and Table 1b, showing

the average performance metrics across the test set samples. Since various combinations of graph architectures and pre-trained language models were tested for node embeddings, we report only the performance of the best-performing combination in each table. WALLE generally outperforms other methods across all metrics on both dataset splits. As previously noted, existing methods do not evaluate interaction prediction; therefore, the baseline performance for bipartite link prediction is provided in Table S8 for future reference.

Table 1: Performance on test set from dataset split by epitope to antigen surface ratio and epitope groups.

(a) Performance on dataset split by epitope to antigen surface ratio.

| Method | MCC | Precision | Recall | AUCROC | F1 |
|---|---|---|---|---|---|
| WALLE | **0.305** (0.023) | **0.308** (0.019) | **0.516** (0.028) | **0.695** (0.015) | **0.357** (0.021) |
| EpiPred | 0.029 (0.018) | 0.122 (0.014) | 0.180 (0.019) | — | 0.142 (0.016) |
| ESMFold | 0.028 (0.010) | 0.137 (0.019) | 0.043 (0.006) | — | 0.060 (0.008) |
| ESMBind | 0.016 (0.008) | 0.106 (0.012) | 0.121 (0.014) | 0.506 (0.004) | 0.090 (0.009) |
| MaSIF-site | 0.037 (0.012) | 0.125 (0.015) | 0.183 (0.017) | — | 0.114 (0.011) |

(b) Performance on dataset split by epitope groups.

| Method | MCC | Precision | Recall | AUCROC | F1 |
|---|---|---|---|---|---|
| WALLE | **0.152** (0.019) | **0.207** (0.020) | **0.299** (0.025) | **0.596** (0.012) | **0.204** (0.018) |
| EpiPred | -0.006 (0.015) | 0.089 (0.011) | 0.158 (0.019) | — | 0.112 (0.014) |
| ESMFold | 0.018 (0.010) | 0.113 (0.019) | 0.034 (0.007) | — | 0.046 (0.009) |
| ESMBind | 0.002 (0.008) | 0.082 (0.011) | 0.076 (0.011) | 0.500 (0.004) | 0.064 (0.008) |
| MaSIF-site | 0.046 (0.014) | 0.164 (0.020) | 0.174 (0.015) | — | 0.128 (0.012) |

**MCC**: Matthews Correlation Coefficient; **AUCROC**: Area Under the Receiver Operating Characteristic Curve; **F1**: F1 score. Standard errors are included in the parentheses. We omitted the results of EpiPred, ESMFold and MaSIF-site for AUCROC. For EpiPred and ESMFold, the interface residues are determined from the predicted structures by these methods such that the predicted values are binary and not comparable to other methods; As for MaSIF-site, it outputs the probability of mesh vertices instead of node probabilities and epitopes are determined as residues close to mesh vertices with probability greater than 0.7.

Table 2: Summary of Features Used in Benchmarking Methods.

| | Antibody | Structure | PLM | Graph |
|---|---|---|---|---|
| WALLE | ✓ | ✓ | ✓ | ✓ |
| EpiPred | ✓ | ✓ | × | ✓ |
| ESMFold | ✓ | × | ✓ | × |
| MaSIF-site | × | ✓ | × | ✓ |
| ESMBind | × | × | ✓ | × |

Antibody: Antibody is taken into consideration when predicting epitope nodes;
Structure: Topological information from protein structures;
PLM: Representation from Protein Language Models;
Graph: Graph representation of protein structures.

Additionally, we benchmarked AlphaFold2-Multimer (AF2M) version 2.3 on the epitope prediction task due to its growing use in complex structure prediction. A new subset of 76 AsEP complexes, excluded from the AF2M training set, was curated for this purpose. Details on the filtering method, the specific AsEP files in this subset, and AF2M performance results can be found in Appendix A.9 and Table S5. While AF2M achieves an MCC of 0.262, its performance could be further improved. Additionally, the average runtime per antibody-antigen pair is 1.66 hours, which is not optimal for epitope scanning, especially given that all other benchmarked methods here can make predictions within seconds.

**Hybrid vs structural only and unsturctured only**   We carried out ablation studies (Appendix C) to investigate the impact of different components of WALLE. Specifically, we investigated the combination of GCN with AntiBERTy and ESM-35M for antibody and antigen embeddings. When we replace GCN layers with fully connected layers, the MCC metric decreases by approximately 39.8%, suggesting that GNN layers contribute to performance. This is related to the fact that the interaction between a pair of protein structures depends on the spatial arrangement of the residues (see Reis et al. (2022)). The interface polar bonds, a major source of antibody specificity, tend to shield interface hydrophobic clusters. The PLM embeddings also contribute to performance, as performance drops over 62.4% when they are replaced with one-hot or BLOSUM62 embeddings. Finally, we investigated whether the choice of PLM affects the model's performance. We found that using AntiBERTy and ESM2 embeddings for antibodies and antigens performed slightly better than using ESM2 embeddings for both antibodies and antigens. This suggests that the choice of the protein language model may impact the model's performance, but a model like ESM2, which is trained on general protein sequences, may contain sufficient information for the epitope prediction task.

WALLE's model architecture is different from existing approaches in that it combines a graph-based method with pre-trained protein language model embeddings. This integration of structural and unstructured (sequential) information leads to a more comprehensive and information-rich approach to epitope prediction. Unlike WALLE, prior graph-based methods, such as PECAN (Pittala & Bailey-Kellogg, 2020) and EPMP (Vecchio et al., 2021), relied on manually engineered features for node embeddings rather than leveraging pre-trained language models. WALLE's node embedding update strategy is similar to that of PECAN, where antibody and antigen graphs are updated separately through distinct graph layers. However, EPMP employs GAT layers to jointly update both antibody and antigen embeddings, incorporating a residual connection to add the updated embeddings back to each graph. WALLE also differs in its task formulation and decoder. While PECAN and EPMP primarily focus on node classification tasks, WALLE predicts bipartite link labels and subsequently aggregates them to obtain node labels. Furthermore, WALLE enhances protein language model embeddings with molecular structures, which is different from methods that rely exclusively on sequence-based information, such as ESMBind and ESMFold, or methods that only use structural information, such as EpiPred and MaSIF-site.

**Generalization to unseen epitopes** While WALLE outperforms other methods in the *epitope group* dataset split setting, its performance degenerated considerably from the first dataset split setting. This suggests that WALLE is likely biased toward the epitopes in the training set and does not generalize well to unseen epitopes. The performance of the other four methods is not ideal for this task as well. To improve the performance of epitope prediction, we believe this would require a more comprehensive dataset and a more sophisticated model architecture, for example using inductive GNNs (Teru et al., 2020; Zhu et al., 2021; Chen et al., 2022) or pretraining protein language models with active forgetting so that their feature extraction can generalize to unseen epitope, a trick recently proposed to help generalization to unseen languages (Chen et al., 2023). We invite researchers to examine the generalization issue and bring their insights and algorithms to address this issue.

**Edge features** In terms of structure representation, we only used a simple invariant edge feature, the distance matrix, to capture the neighborhood information of each residue. This topological descriptor already performs better than other methods that use sequence-based features. For future work, more edge features can be incorporated to enrich the graph representation, in addition to the invariant edge features used in this work, such as inter-residue distances and edge types used in GearNet (Zhang et al., 2023), and equivariant features, such as rotational and orientational relationships between residues as used in abdockgen (Jin et al., 2022). The incoporation of edge features will invite testing of advanced graph learning algorithms for multi-relational graphs (Nickel et al., 2011; Bordes et al., 2013; Schlichtkrull et al., 2018; Murphy et al., 2019; Chen et al., 2021).

**Dataset** We plan to extend our work to include more types of antibodies. Currently, our dataset focuses solely on conventional antibodies composed of heavy-chain and light-chain variable domains. However, there is a growing interest in novel antibody formats, such as nanobodies, which are single-variable-domain antibodies derived from camelids. These will be included in future releases of AsEP. Additionally, given the abundance of general non-antibody-antigen complexes, pre-training on such complexes could be beneficial in providing the downstream epitope prediction models with a foundational understanding of protein interface, as suggested by PECAN (Pittala & Bailey-Kellogg, 2020). Further finetuning on antibody-antigen complexes would then allow the model to capture the unique characteristics of antibody interfaces, such as specific amino acid compositions and interaction

types, that distinguish them from general protein interfaces (Kringelum et al., 2013; Ofran et al., 2008). We leave expansions of our dataset to incorporate such diverse data, pretraining over general proteins and finetuning vertically for antibody-antigen complexes as our future work.

# 7 Conclusion

In this work, we proposed AsEP, a novel benchmarking dataset for the epitope prediction task and the first dataset to cluster antibody-antigen complexes by epitopes. We also introduced WALLE, a model that combines pretrained protein language models with graph neural networks to capture both amino acid contextual and geometric information. We benchmarked WALLE alongside four other methods, demonstrating that it outperforms existing methods on both tasks, though there is still plenty of room for enhancement, especially on the unseen epitopes. Our results suggest that such integration of both structural modeling and unstructured (sequential) modeling improves epitope prediction performance. We discussed future work that could enhance model performance, including enriching edge features for better structural representation, refining model architecture, exploring transfer learning from general protein complexes, and expanding the dataset to include emerging antibody types. This work serves as our starting point for advancing research in antibody-specific epitope prediction.

# 8 Acknowledgments

CL was part-funded by a UCL Centre for Digital Innovation Amazon Web Services (AWS) Scholarship; LD was funded by an ISMB MRC iCASE Studentship (MR/R015759/1). We would also like to thank the NeurIPS reviewers for their valuable feedback and suggestions, which helped us improve the clarity and quality of this work.

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

# A    Appendix-A

## A.1    Related work

**Comparison of Previous Datasets** We would like to highlight our dataset, AsEP, is the largest curated AbAg benchmarking dataset to date. Existing ones either focus on general protein-protein complexes designed to develop general docking methods or are way smaller than AsEP if designed for AbAg interaction research. We summarized the sizes of existing datasets in the following table.

Table S1: Comparison of Dataset Sizes Across Different Methods

| Method | Dataset Size |
|---|---|
| WALLE (AsEP) | 1723 AbAg complexes |
| Wang et al. 2022 (Wang et al., 2022) | 258 AbAg complexes |
| SAGERank (Sun et al., 2023) | 287 AbAg complexes |
| CSM-AB (Myung et al., 2021) | 472 AbAg complexes |
| Bepipred3.0 (Clifford et al., 2022) | 582 AbAg complexes |

In the work by Wang et al. (2022) the dataset from Zhao et al. (2018) was used, which included 257 antibody-antigen complexes, encompassing both VHVL and VHH antibodies. However, the primary focus of this research is on epitope prediction using the antigen as the input. Consequently, antibodies were not included in the predictive models, rendering the dataset unsuitable for antibody-specific epitope prediction.

For CSM-AB, as described by their supplementary information, the dataset contains 472 antibody-antigen structures including 375 Fab, 82 Nanobody and 12 scFv (Myung et al., 2021). These structures were collected from PDB, identified using Chothia annotation as in (Dunbar & Deane, 2015). The authors did not describe the procedure of any further filtering steps. We assumed they included all available structures at that time with available protein-protein binding affinity information from PDBbind since the study aims to predict binding affinity.

The authors of SAGERank formed a dataset composed of 287 antibody-antigen complexes filtered by sequence identity at 95% (Sun et al., 2023) . While the authors did not explicitly mention the antibody types, we infer from the results that these are also Fab antibodies that include both VH and VL domains. The dataset was composed mainly for docking pose ranking output by MegaDock and did not include interface clustering, i.e. epitope grouping.

SCEptRe by  Mahajan et al. (2019) is a related dataset that keeps a weekly updated collection of 3D complexes of epitope and receptor pairs, for example, antibody-antigen, TCR-pMHC, and MHC-ligand complexes derived from the Immune Epitope Database (IEDB). Our approach for clustering antibody-antigen complexes regarding their epitopes is similar to theirs, with the difference in the clustering strategy. We cluster by antigen, then epitope group, and we allow mutated amino acids in the same epitope region because we turn the epitope sites into columns in the multiple sequence alignment. In contrast, SCEptRe clusters by antibody and then compares epitope similarity by epitope conformation using atom-pair distances via PocketMatch (Yeturu & Chandra, 2008), which is beneficial for comparing the function of various paratopes but is less suitable for our task of predicting epitope residues.

**Sequence-based epitope predictor** We also tested purely sequence-based epitope prediction tool, for example, Bepipred3.0 (Clifford et al., 2022) on our dataset. Bepipred3.0 uses ESM2 model, `esm2_t33_650M_UR50D` to generate sequence embeddings and was trained on a smaller dataset of 582 antibody-antigen structures and evaluated on 15 antibody-antigen complexes. The authors provided a relatively larger evaluation of linear B-cell epitopes derived from the Immune Epitope Database and reported an AUC-ROC of 0.693 on the top 10% of the predicted epitopes. We tested Bepipred3.0 on our dataset and found its performance degenerates significantly, as shown in the table below. This is not surprising because linear epitopes are consecutive positions in an antigen sequence, and this task fits better with language model design. Additionally, as pointed out by the authors, approximately 90% of epitopes (B-cell) fall into the conformational category (Clifford et al., 2022), which highlights the importance of the present benchmark dataset composed of conformational epitopes derived from filtered antibody-antigen  structures. We believe these results underline the

findings in our paper, showing that large language models alone, even if specialized for antibody-antigen interactions, do not encompass all the relevant information needed for epitope prediction.

| **Confidence** Threshold | **Top 10%** | **Top 30%** | **Top 50%** | **Top 70%** | **Top 90%** |
|---|---|---|---|---|---|
| AUC | 0.693392 | 0.693392 | 0.693392 | 0.693392 | 0.693392 |
| Balanced Accuracy | 0.573132 | 0.636365 | 0.638755 | 0.604556 | 0.542274 |
| MCC | 0.109817 | 0.140183 | 0.134689 | 0.113372 | 0.071876 |
| Precision-Recall AUC | 0.176429 | 0.176429 | 0.176429 | 0.176429 | 0.176429 |
| Accuracy | 0.850178 | 0.701202 | 0.536947 | 0.362489 | 0.179051 |
| Precision | 0.169202 | 0.141361 | 0.120547 | 0.104286 | 0.090441 |
| Recall | 0.236760 | 0.553204 | 0.756607 | 0.892628 | 0.977723 |
| F1-Score | 0.173370 | 0.208366 | 0.197153 | 0.179294 | 0.160151 |

Table S2: Bepipred3.0 results for the presented AsEP dataset. The distance cutoff was changed to 4.0Å, as this is the threshold used by Bepipred3.0. Results are shown for five confidence thresholds as described in the BepiPred-3.0 paper. Across all stringency settings and metrics, Bepipred scored lower than Walle. Furthermore, it is possible that some of the structures within the dataset are contained within the Bepipred3.0 dataset, artificially increasing scores.

## A.2 Evaluation

In this work, we focus on the epitope prediction task. We evaluate the performance of each method using consistent metrics. Matthew's Correlation Coefficient (MCC) is highly recommended for binary classification assessments (Matthews, 1975) and is especially advocated for its ability to provide equal weighting to all four values in the confusion matrix, making it a more informative metric about the classifier's performance at a given threshold than other metrics (Chicco & Jurman, 2020, 2023). We encourage the community to adopt MCC for the epitope prediction task as it takes into account true and false positives, as well as true and false negatives, offering a comprehensive measure of the performance. It is considered superior to the AUC-ROC, which is evaluated over all thresholds. For consistency, we also included Precision and Recall from prior studies EpiPred (Krawczyk et al., 2014) and PECAN (Pittala & Bailey-Kellogg, 2020), and we added Area Under the Receiver Operating Characteristic Curve (AUC-ROC) and F1 score, both are typical binary classification metrics. For methods that predict antibody-antigen complex structures, we determine the epitopes using the same distance criterion as aforementioned.

## A.3 Steps to build antibody-antigen complex dataset

We sourced our initial dataset from AbDb (version dated September 26, 2022), containing 11,767 antibody files originally collected from the Protein Data Bank (PDB). We collected complexes numbered in Martin scheme (Raghavan & Martin, 2008) and used AbM CDR definition (Martin et al., 1991) to identify CDR residues from the heavy and light chains of antibodies.

We extracted antibody-antigen complexes that met the following criteria: (1) both VH and VL domains are present in the antibody; (2) the antigen is a single-chain protein consisting of at least 50 amino acids; and (3) there are no unresolved CDR residues, yielding 4,081 files.

To deduplicate complexes, we used MMseqs2 (Steinegger & Söding, 2017) to cluster the complexes by heavy and light chains in antibodies and antigen sequences. We used the *easy-linclust* mode with the *–cov-mode 0* option to cluster sequences; we used the default setting for coverage of aligned cutoff at 80%; we used different *–min-seq-id* cutoffs for antibodies and antigens because the antibody framework regions are more conserved than the CDR regions. We cluster heavy and light chains at *–min-seq-id* cutoff of 100% and 70%, respectively. We retained only one representative file for each unique set of identifiers, leading to a refined dataset of 1,725 files.

Two additional files were manually removed. File *6jmr_1P* was removed because its CDR residues are masked with 'UNK' labels and the residue identities are unknown; file *7sgm_0P* was removed because of a non-canonical residue 'DV7' in its CDR-L3 loop.

The final dataset consists of 1,723 antibody-antigen complexes.

## A.4 Pipeline to build graph dataset from AbDb

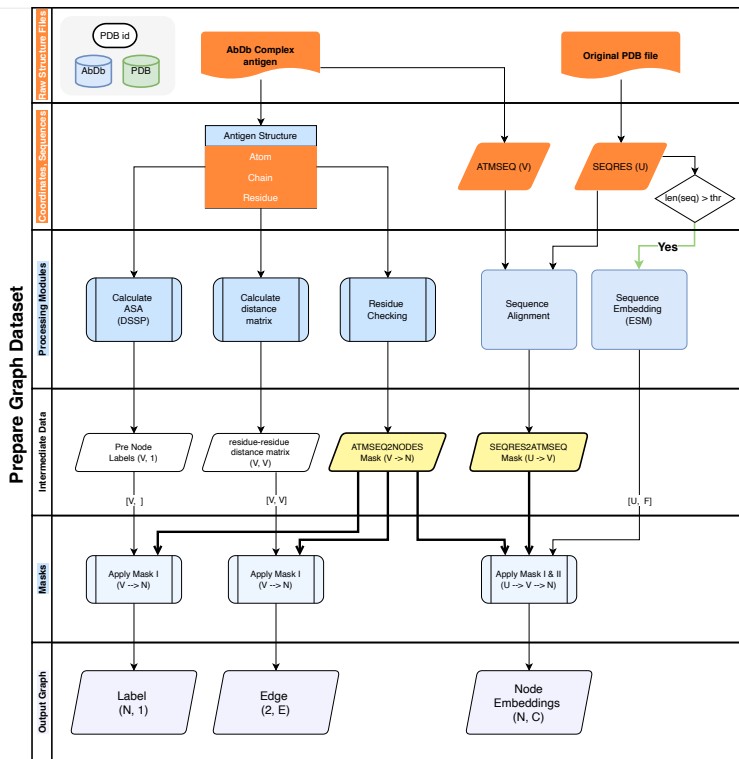

Figure S1: Pipeline to convert an antibody-antigen complex structure into a graph representation.

- **Row 1**: given an AbAg complex PDB ID, retrieve 'AbAg complex' from AbDb and 'raw structure' file from PDB as input, 'AbDb complex antigen' and 'Original PDB file' in the top lane.
- **Row 2**: They are then parsed as hierarchical coordinates (Antigen Structure), and extract ATMSEQ and SEQRES sequences.
- **Row 3**: these are then passed to a set of in-house modules for calculating solvent access surface area (ASA), distance matrix, and filtering out problematic residues, which generates an ATMSEQ2NODES mask. The sequence alignment module aligns ATMSEQ with SEQRES sequences to generate a mask mapping from SEQRES to ATMSEQ. The Sequence Embedding module passes SEQERS through the ESM module to generate embeddings. ESM requires input sequence length and therefore filters out sequences longer than 1021 amino acids.
- **Row 4**: holds intermediate data that we apply masks to generate graph data in **Row 5**.
- **Row 6**: Apply the masks to map SEQRES node embeddings to nodes in the graphs and calculate the edges between the graph nodes.

$U$, $V$ and $N$ denote the number of residues in the *SEQRES* sequence, *ATMSEQ* sequence and the graph, respectively. *thr* (at 50 residues) is the cutoff for antigen *SEQRES* length. We only include antigen sequences with lengths of at least 50 residues. *SEQRES* and *ATMSEQ* are two different sequence representations of a protein structure. *SEQRES* is the sequence of residues in the protein chain as defined in the header section of a PDB file, and it is the complete sequence of the protein chain. *ATMSEQ* is the sequence of residues in the protein chain as defined in the ATOM section of a PDB file. In other words, it is read from the structure, and any residues in a PDB file are not resolved due to experimental issues that will be missing in the *ATMSEQ* sequence. Since we are building graph representations using structures, we used *ATMSEQ*. However, the input to the language models

require a complete sequence, therefore we used *SEQRES* to generate node embeddings, and mapped the node embeddings to the graph nodes. We performed two pairwise sequence alignments to map such embeddings to graph vertices for a protein chain with Clustal Omega (Sievers et al., 2011). We first align the SEQRES sequence with the atom sequence (residues collected from the ATOM records in a PDB file) and assign residue embeddings to matched residues. Because we excluded buried residues from the graph, we aligned the sequence formed by the filtered graph vertices with the atom sequence to assign residue embeddings to vertices. *ASA* is the solvent-accessible surface area of a residue. If a residue has an ASA value of zero, it is considered buried and will be removed from the graph.

## A.5 Dataset split

Table S3: Distribution of epitope to antigen surface nodes in each set.

| Epi/Surf | Training | Validation/Test |
|---|---|---|
| 0, 5% | 320 (23% ) | 40 (24%) |
| 5% , 10% | 483 (35% ) | 60 (35%) |
| 10%, 15% | 305 (22% ) | 38 (22%) |
| 15%, 20% | 193 (14% ) | 24 (14%) |
| 20%, 25% | 53 (4% ) | 6 (4% ) |
| 25%, 30% | 19 (1% ) | 2 (1% ) |
| 30%, 35% | 8 (0.6%) | 0 (- ) |
| 35%, 40% | 2 (0.1%) | 0 (- ) |
| sum | 1383 | 170 |

## A.6 Implementation details

**Exploratory Data Analysis** We performed exploratory data analysis on the training dataset to understand the distribution of the number of residue-residue contacts in the antibody-antigen interface. We found that the number of contacts is approximately normally distributed with a mean of 43.42 and a standard deviation of 11.22 (Figure S2). We used this information to set the regularizer in the loss function to penalize the model for predicting too many or too few positive edges.

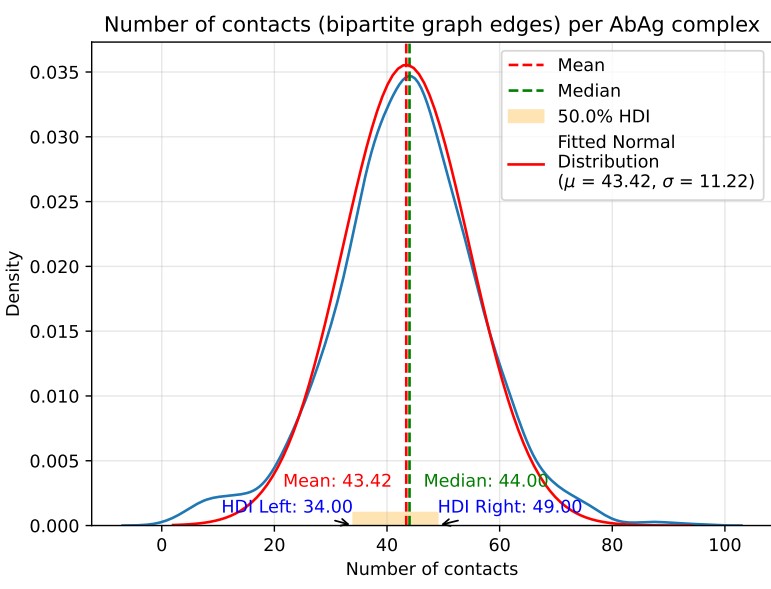

Figure S2: Blue line: distribution of the number of residue-residue contacts in antibody-antigen interface across the dataset with a mean and median of 43.27 and 43.00, respectively. Red line: fitted normal distribution with mean and standard deviation of 43.27 and 10.80, respectively.

**Loss function** Our loss function is a weighted sum of two parts: a binary cross-entropy loss for the bipartite graph linkage reconstruction and a regularizer for the number of positive edges in the reconstructed bipartite graph.

$$\text{Loss} = \mathcal{L}_r + \lambda \left| \sum^N \hat{e} - c \right| \tag{3}$$

$$\mathcal{L}_r = -\frac{1}{N} \sum_{i=1}^{N} \left( w_{\text{pos}} \cdot y_e \cdot \log(\hat{y_e}) + w_{\text{neg}} \cdot (1 - y_e) \cdot \log(1 - \hat{y_e}) \right) \tag{4}$$

The binary cross-entropy loss $\mathcal{L}_r$ is weighted by $w_{\text{pos}}$ and $w_{\text{neg}}$ for positive and negative edges, respectively. During hyperparameter tuning, we kept $w_{\text{neg}}$ fixed at $1.0$ and tuned $w_{\text{pos}}$. $N$ is the total number of edges in the bipartite graph, $y_e$ denotes the true label of edge $e$, and $\hat{y_e}$ denotes the predicted probability of edge $e$. The regularizer $\left| \sum^N \hat{e} - c \right|$ is the L1 norm of the difference between the sum of the predicted probabilities of all edges of the reconstructed bipartite graph and the mean positive edges in the training set, i.e., $c$ set to $43$. This aims to prevent an overly high false positive rate, given the fact that the number of positive edges is far less than positive edges. The regularizer weight $\lambda$ is tuned during hyperparameter tuning.

**Hyperparameters** We carried out hyperparameter search within a predefined space that included:

- **Weights for positive edges** in bipartite graph reconstruction loss, sampled uniformly between 50 and 150.
- **Weights for the sum of bipartite graph positive links**, where values were drawn from a log-uniform distribution spanning $1e - 7$ to $1e - 4$.
- **Edge cutoff (x)**, defining an epitope node as any antigen node with more than x edges, with x sampled following a normal distribution with a mean of 3 and a standard deviation of 1.
- **Number of graph convolutional layers** in the encoder, we tested using 2 and 3 layers.
- **Decoder type** was varied between two configurations:
    - A fully connected layer, equipped with a bias term and a dropout rate of 0.1.
    - An inner product decoder.

**Computing resources** All experiments were performed on a single Linux machine (Ubuntu 22.04) with one NVIDIA RTX3090 GPU card, and it took on average 3 hours for a single hyper-parameter sweeping experiment.

### A.7 Antibody-antigen complex examples

To compare the epitopes of antibodies in AsEP, we first clustered these complexes by antigen sequences to group together antibodies targeting the same antigen via MMseqs2 (Steinegger & Söding, 2017). Specifically, we ran MMseqs2 on antigen SEQRES sequences and using the following setup:

- `easy-linclust` mode
- `cov-mode` set to 0 with the default coverage of 80%: this means a sequence is considered a cluster member if it aligns at least 80% of its length with the cluster representative;
- `min-seq-id` set to `0.7`: this means a sequence is considered a cluster member if it shares at least 70% sequence identity with the cluster representative.

We encourage the reader to refer to the MMseqs2 documentation `https://github.com/soedinglab/mmseqs2/wiki` for more details on the parameters used.

We then identify epitopes using a distance cut-off of 4.5 Å. An antigen residue is identified as epitope if any of its heavy atoms are located within 4.5 Å of any heavy atoms from the antibody.

To compare epitopes of antibodies sharing the same antigen cluster, we aligned the antigen SEQRES sequences using Clustal Omega (Sievers et al., 2011) (download from: `http://www.clustal.org/omega/`) to obtain a Multiple Sequence Alignment (MSA). Epitopes are mapped to and denoted as the MSA column indices. The epitope similarity between a pair of epitopes is then calculated as

the fraction of identical columns. Two epitopes are identified as identical if they share over 0.7 of identical columns.

Table S4: Antibody-Antigen Complex Examples

(a) Antigen Group Information

| abdbid | repr | size | epitope_group |
|--------|------|------|---------------|
| 7eam_1P | 7sn2_0P | 183 | 0 |
| 5kvf_0P | 5kvd_0P | 9 | 1 |
| 5kvg_0P | 5kvd_0P | 9 | 2 |

(b) CDR Sequences (Heavy Chain)

| abdbid | H1 | H2 | H3 |
|--------|-----|-----|-----|
| 7eam_1P | GFNIKDTYIH | RIDPGDGDTE | FYDYVDYGMDY |
| 5kvf_0P | GYTFTSSWMH | MIHPNSGSTN | YYYDYDGMDY |
| 5kvg_0P | GYTFTSYGIS | VIYPRSGNTY | ENYGSVY |

(c) CDR Sequences (Light Chain)

| abdbid | L1 | L2 | L3 |
|--------|-----|-----|-----|
| 7zf4_1P | RASGNIHNYLA | NAKTLAD | QHFWSTPPWT |
| 7zbu_0P | KSSQSLLYSSNQKNYLA | WASTRES | QQYYTYPYT |
| 7xxl_0P | KASQNVGTAVA | SASNRYT | QQFSSYPYT |

(d) Structure Titles

| abdbid | resolution | title | repr_title |
|--------|------------|-------|------------|
| 7zf4_1P | 1.4 | immune complex of SARS-CoV-2 RBD and cross-neutralizing antibody 7D6 | SARS-CoV-2 Omicron variant spike protein in complex with Fab XGv265 |
| 7zbu_0P | 1.4 | Zika specific antibody, ZV-64, bound to ZIKA envelope DIII | Cryo-EM structure of zika virus complexed with Fab C10 at pH 8.0 |
| 7xxl_0P | 1.4 | Zika specific antibody, ZV-67, bound to ZIKA envelope DIII | Cryo-EM structure of zika virus complexed with Fab C10 at pH 8.0 |

Here we provide three example antibody-antigen complexes from the same antigen group, meaning the antigen sequences from each member complex share sequence identity of the aligned region at least 70%. Due to space limitation, we have broken the rows into four parts: Antigen group information, CDR sequences, and structure titles.

- **abdbid**: AbDb ID of the group member;
- **repr**: AbDb ID of the antigen representative;
- **size**: the number of complexes in the group;
- **epitope_group**: A categorical identifier of the epitope group the antibody-antigen complex belongs to;
- **H1**, **H2**, **H3**, **L1**, **L2**, **L3**: CDR sequences of the heavy and light chains;
- **resolution**: Structure resolution;
- **title**: Structure title of the member;
- **repr_title**: Structure title of the antigen representative.

### A.8 Multi-epitope Antigens

We include 641 unique antigens and 973 epitope groups in our dataset. Figure S3 shows two examples of multi-epitope antigens in our dataset, hen egg white lysozyme (Figure S3a) and spike protein (Figure S3b). Specifically, there are 52 and 64 distinct antibodies in our dataset that bind to hen egg white lysozyme and spike protein, respectively. For visual clarity, we only show five and sixteen antibodies in Figure S3a and Figure S3b.

We can see that different antibodies bind to different locations on the same antigen. Details of all epitope groups are provided in the Supplementary Table `SI-AsEP-entries.csv` with annotation provided in Appendix A.7.

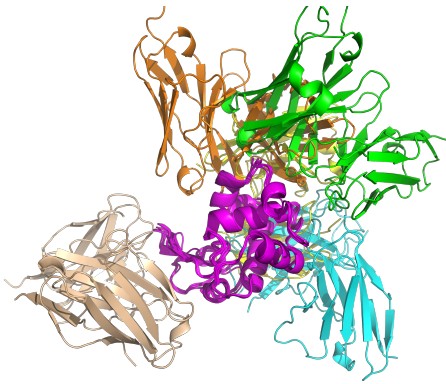

(a) Five different antibodies bound to hen egg white lysozyme. Complexes are superimposed on the antigen structure (magenta). AbDb IDs of the complexes and their color: 1g7i_0P (green), 2yss_0P (cyan), 1dzb_1P (yellow), 4tsb_0P (orange), 2iff_0P (wheat). Antigens are colored in magenta.

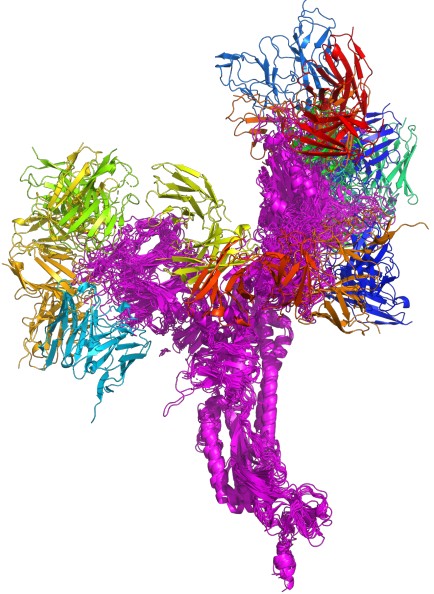

(b) Sixteen different antibodies bound to coronavirus spike protein. Complexes are superimposed on the antigen structure (magenta) and antibodies are in different colors. AbDb IDs of the complexes: 7k8s_0P, 7m7w_1P, 7d0b_0P, 7dzy_0P, 7ey5_1P, 7jv4_0P, 7k8v_1P, 7kn4_1P, 7lqw_0P, 7n8i_0P, 7q9i_0P, 7rq6_0P, 7s0e_0P, 7upl_1P, 7wk8_0P, 7wpd_0P.

Figure S3: Examples of different antibodies binding to the same antigen.

## A.9 AsEp subset for Benchmarking AlphaFold 2.3 Multimer

As documented in the AlphaFold technical notes (`https://github.com/google-deepmind/alphafold/blob/main/docs/technical_note_v2.3.0.md`, the AlphaFold version 2.3 Multimer (AF2M) model's training cutoff date is September 30, 2021. To assess AF2M's generalizability to novel epitopes not present in the training set, we identified unique epitopes by filtering based on this cutoff date. First, we retrieved all antibody-antigen complex structures from AbDb (snapshot: 2022 September 26) and clustered the antigen sequences using MMseqs2 with the following configuration. The resulting groups are referred to as *antigen clusters*:

```
easy-cluster
--min-seq-id=0.9  # min sequence identify
--cov-mode=0      # coverage includes both query (cluster representative)
                  # and target (cluster member) sequence
--coverage=0.8    # min coverage
```

Next, we obtained the release date of each AbDb file, cross-referencing with the Protein Data Bank (PDB), and retained only those files in AsEP that were released after the training cutoff date. These files were mapped to their respective *antigen clusters*.

Finally, we kept only those structures where all cluster members were released after the cutoff date. This filtering process yielded the following 76 structures:

```
AbDb files:
7pg8_1P 7lk4_3P 7rd5_1P 7tzh_1P 7zyi_0P 7pnq_0P 7rfp_1P 7lr4_0P 7tuf_1P
7pnm_0P 7rk2_1P 7seg_1P 7nx3_1P 7rk1_1P 7amq_0P 7sk9_0P 7yrf_0P 7teq_0P
7usl_0P 7vyt_1P 7tpj_0P 7ucg_2P 7lo7_0P 7l7r_1P 7vaf_0P 7t0r_0P 7ttm_0P
7lr3_1P 7tuy_0P 7a3o_0P 7wk8_0P 7ue9_0P 7stz_1P 7fbi_0P 7sfv_0P 7tdm_0P
7xw6_0P 7fbi_1P 7dce_0P 7q6c_0P 7s11_2P 7rxd_0P 7w71_1P 7sbg_0P 7zwf_0P
7r9d_0P 7oh1_0P 7po5_0P 7qu1_0P 7a3p_0P 7ttx_0P 7ura_0P 7zwm_2P 7sjn_0P
7xw7_0P 7a3t_0P 7tzh_0P 7l7r_0P 7a0w_0P 7u8m_3P 7lo8_0P 7zwm_3P 7mnl_1P
7n0a_0P 7bh8_1P 7fci_0P 8dke_0P 7daa_0P 7tug_0P 7mrz_0P 7tdn_0P 7dkj_0P
7amr_0P 7zxf_0P 7v61_1P 7teb_0P
```

## A.10 AlphaFold2.3 Multimer performance

We used the top one-ranked model generated by AlphaFold2.3 Multimer in the performance calculation in Table S5. See Appendix A.9 for the list of AbDb files used for this benchmark. The average runtime, including both constructing MSA and predicting structures, is 1.66 hours (standard deviation: 1.19 hours, median: 1.27 hours).

Table S5: AlphaFold2 Multimer performance on novel epitopes after its training cutoff date, 2021 September 30

| Method | MCC | Precision | Recall | F1 |
|--------|-----|-----------|--------|-----|
| AF2M | 0.262 (0.039) | 0.276 (0.033) | 0.396 (0.043) | 0.313 (0.036) |

## A.11 Metrics definition

$$\text{MCC} = \frac{(TP \times TN - FP \times FN)}{\sqrt{(TP+FP)(TP+FN)(TN+FP)(TN+FN)}}$$

$$\text{Precision} = \frac{TP}{TP+FP}$$

$$\text{Recall} = \frac{TP}{TP+FN}$$

$$F1 = \frac{2 \times \text{Precision} \times \text{Recall}}{\text{Precision} + \text{Recall}}$$

TP: True Positive

FP: False Positive

TN: True Negative

FN: False Negative

## A.12   Fine-tuning ESMBind on AsEP

The performance reported in the main text for ESMBind is derived by fine-tuning ESM2 on general protein binding sites. We performed a further fine-tuning experiment, fine-tuning it on the presented AsEP dataset and evaluating it on the AsEP test set to enable a more direct comparison of ESMBind to WALLE. Fine-tuning of ESMBind on AsEP was done using the Low-Rank Adaptation method (Hu et al., 2021).

Table S6: Performance Metrics

| Metric | Value |
|---|---|
| MCC | **0.103923** |
| Accuracy | 0.504478 |
| AUC-ROC | 0.584497 |
| Precision | 0.128934 |
| Recall | 0.707731 |
| F1 | 0.213829 |

# B Appendix-B: Evaluating WALLE with Different Graph Architectures

## B.1 Performance of WALLE variants

In this section, we included WALLE's performance across various graph neural network (GNN) architectures, including GAT, GCN, and GraphSAGE. The results are organized into subsections based on two dataset split strategies: by epitope-to-antigen surface ratio and by epitope group. In Table S7 and Table S8, the protein language models (PLM) used for generating antibody and antigen node embeddings include AntiBERTy, ESM2-35M and ESM2-650M. AntiBERTy is exclusively for generating antibody node embeddings. ESM2-35M and ESM2-650M stand for *esm2_t12_35M_UR50D* and *esm2_t33_650M_UR50D*, respectively. The performance values are shown as mean metrics with standard errors provided in parentheses.

Table S7: Performance on epitope prediction

(a) Performance on dataset split by epitope to antigen surface ratio

| Method | antibody PLM | antigen PLM | MCC | Precision | Recall | AUCROC | F1 |
|---|---|---|---|---|---|---|---|
| GAT | AntiBERTy | ESM2-35M | 0.266 (0.023) | 0.287 (0.020) | 0.490 (0.027) | 0.669 (0.014) | 0.325 (0.020) |
| GCN | AntiBERTy | ESM2-35M | 0.264 (0.021) | 0.258 (0.017) | **0.534** (0.027) | 0.680 (0.014) | 0.322 (0.019) |
| GraphSAGE | AntiBERTy | ESM2-35M | 0.288 (0.022) | 0.296 (0.019) | 0.521 (0.027) | 0.682 (0.014) | 0.350 (0.020) |
| GAT | ESM2-35M | ESM2-35M | 0.254 (0.021) | 0.258 (0.016) | **0.534** (0.027) | 0.669 (0.014) | 0.322 (0.018) |
| GCN | ESM2-35M | ESM2-35M | 0.246 (0.019) | 0.234 (0.014) | 0.607 (0.025) | 0.681 (0.013) | 0.310 (0.016) |
| GraphSAGE | ESM2-35M | ESM2-35M | 0.264 (0.021) | 0.269 (0.017) | 0.528 (0.027) | 0.674 (0.013) | 0.327 (0.019) |
| GAT | AntiBERTy | ESM2-650M | 0.279 (0.024) | **0.315** (0.022) | 0.437 (0.028) | 0.664 (0.014) | 0.333 (0.022) |
| GCN | AntiBERTy | ESM2-650M | **0.305** (0.023) | 0.308 (0.019) | 0.516 (0.028) | **0.695** (0.015) | **0.357** (0.021) |
| GraphSAGE | AntiBERTy | ESM2-650M | 0.294 (0.024) | 0.301 (0.020) | 0.506 (0.029) | 0.684 (0.014) | 0.351 (0.022) |
| GAT | ESM2-650M | ESM2-650M | 0.243 (0.021) | 0.291 (0.021) | 0.434 (0.028) | 0.648 (0.014) | 0.297 (0.018) |
| GCN | ESM2-650M | ESM2-650M | 0.264 (0.022) | 0.276 (0.019) | 0.509 (0.028) | 0.672 (0.014) | 0.326 (0.019) |
| GraphSAGE | ESM2-650M | ESM2-650M | 0.270 (0.021) | 0.269 (0.016) | 0.533 (0.027) | 0.678 (0.013) | 0.336 (0.018) |

(b) Performance on dataset split by epitope groups

| Method | antibody PLM | antigen PLM | MCC | Precision | Recall | AUCROC | F1 |
|---|---|---|---|---|---|---|---|
| GAT | AntiBERTy | ESM2-35M | 0.085 (0.017) | 0.164 (0.016) | 0.225 (0.021) | 0.550 (0.010) | 0.165 (0.015) |
| GCN | AntiBERTy | ESM2-35M | 0.122 (0.019) | 0.186 (0.018) | 0.267 (0.024) | 0.576 (0.011) | 0.187 (0.017) |
| GraphSAGE | AntiBERTy | ESM2-35M | 0.113 (0.016) | 0.181 (0.015) | 0.259 (0.020) | 0.568 (0.010) | 0.196 (0.015) |
| GAT | ESM2-35M | ESM2-35M | 0.109 (0.016) | 0.175 (0.016) | 0.259 (0.023) | 0.568 (0.010) | 0.177 (0.015) |
| GCN | ESM2-35M | ESM2-35M | 0.123 (0.017) | 0.184 (0.016) | **0.345** (0.025) | 0.583 (0.011) | 0.202 (0.015) |
| GraphSAGE | ESM2-35M | ESM2-35M | 0.113 (0.017) | 0.176 (0.015) | 0.279 (0.022) | 0.572 (0.011) | 0.194 (0.015) |
| GAT | AntiBERTy | ESM2-650M | 0.125 (0.018) | 0.189 (0.017) | 0.251 (0.022) | 0.575 (0.011) | 0.188 (0.016) |
| GCN | AntiBERTy | ESM2-650M | **0.152** (0.019) | **0.207** (0.020) | 0.299 (0.025) | **0.596** (0.012) | **0.204** (0.018) |
| GraphSAGE | AntiBERTy | ESM2-650M | 0.118 (0.018) | 0.206 (0.019) | 0.209 (0.021) | 0.565 (0.011) | 0.176 (0.017) |
| GAT | ESM2-650M | ESM2-650M | 0.103 (0.017) | 0.164 (0.016) | 0.240 (0.021) | 0.566 (0.011) | 0.173 (0.015) |
| GCN | ESM2-650M | ESM2-650M | 0.138 (0.017) | 0.197 (0.018) | 0.271 (0.022) | 0.585 (0.011) | 0.199 (0.016) |
| GraphSAGE | ESM2-650M | ESM2-650M | 0.111 (0.019) | 0.195 (0.019) | 0.212 (0.020) | 0.559 (0.010) | 0.183 (0.017) |

Table S8: Performance on bipartite link prediction

(a) Performance on dataset split by epitope to antigen surface ratio

| Method | antibody PLM | antigen PLM | MCC | Precision | Recall | AUCROC | F1 |
|---|---|---|---|---|---|---|---|
| GAT | AntiBERTy | ESM2-35M | 0.118 (0.010) | 0.047 (0.004) | 0.393 (0.027) | 0.676 (0.014) | 0.079 (0.007) |
| GCN | AntiBERTy | ESM2-35M | 0.114 (0.010) | 0.049 (0.004) | 0.343 (0.027) | 0.657 (0.013) | 0.080 (0.007) |
| GraphSAGE | AntiBERTy | ESM2-35M | 0.122 (0.009) | 0.049 (0.004) | 0.386 (0.025) | 0.675 (0.012) | 0.083 (0.006) |
| GAT | ESM2-35M | ESM2-35M | 0.079 (0.006) | 0.020 (0.001) | **0.493** (0.027) | 0.690 (0.013) | 0.037 (0.003) |
| GCN | ESM2-35M | ESM2-35M | 0.088 (0.008) | 0.035 (0.003) | 0.301 (0.023) | 0.630 (0.011) | 0.060 (0.005) |
| GraphSAGE | ESM2-35M | ESM2-35M | 0.098 (0.008) | 0.038 (0.003) | 0.333 (0.025) | 0.648 (0.012) | 0.064 (0.005) |
| GAT | AntiBERTy | ESM2-650M | 0.121 (0.011) | 0.051 (0.005) | 0.369 (0.029) | 0.670 (0.014) | 0.084 (0.008) |
| GCN | AntiBERTy | ESM2-650M | **0.131** (0.011) | **0.062** (0.005) | 0.354 (0.029) | 0.666 (0.014) | **0.095** (0.008) |
| GraphSAGE | AntiBERTy | ESM2-650M | 0.116 (0.009) | 0.044 (0.003) | 0.373 (0.027) | 0.670 (0.013) | 0.077 (0.006) |
| GAT | ESM2-650M | ESM2-650M | 0.112 (0.009) | 0.036 (0.003) | 0.484 (0.029) | **0.709** (0.014) | 0.064 (0.006) |
| GCN | ESM2-650M | ESM2-650M | 0.119 (0.009) | 0.039 (0.003) | 0.468 (0.029) | 0.707 (0.014) | 0.070 (0.006) |
| GraphSAGE | ESM2-650M | ESM2-650M | 0.112 (0.008) | 0.042 (0.003) | 0.373 (0.025) | 0.668 (0.012) | 0.073 (0.005) |

(b) Performance on dataset split by epitope groups

| Method | antibody PLM | antigen PLM | MCC | Precision | Recall | AUCROC | F1 |
|---|---|---|---|---|---|---|---|
| GAT | AntiBERTy | ESM2-35M | 0.034 (0.005) | 0.020 (0.003) | 0.101 (0.014) | 0.542 (0.007) | 0.029 (0.004) |
| GCN | AntiBERTy | ESM2-35M | 0.048 (0.006) | 0.020 (0.002) | 0.195 (0.022) | 0.581 (0.011) | 0.034 (0.004) |
| GraphSAGE | AntiBERTy | ESM2-35M | 0.055 (0.006) | 0.026 (0.003) | 0.185 (0.019) | 0.578 (0.009) | 0.042 (0.004) |
| GAT | ESM2-35M | ESM2-35M | 0.040 (0.005) | 0.017 (0.003) | 0.222 (0.020) | 0.580 (0.009) | 0.027 (0.003) |
| GCN | ESM2-35M | ESM2-35M | 0.044 (0.005) | 0.019 (0.003) | 0.213 (0.020) | 0.578 (0.009) | 0.031 (0.004) |
| GraphSAGE | ESM2-35M | ESM2-35M | 0.047 (0.006) | 0.022 (0.003) | 0.173 (0.018) | 0.570 (0.009) | 0.035 (0.004) |
| GAT | AntiBERTy | ESM2-650M | 0.054 (0.006) | 0.022 (0.003) | 0.218 (0.022) | 0.591 (0.011) | 0.037 (0.004) |
| GCN | AntiBERTy | ESM2-650M | **0.063** (0.007) | 0.024 (0.003) | **0.265** (0.024) | **0.613** (0.012) | 0.041 (0.004) |
| GraphSAGE | AntiBERTy | ESM2-650M | 0.058 (0.007) | **0.031** (0.003) | 0.174 (0.020) | 0.577 (0.010) | **0.046** (0.005) |
| GAT | ESM2-650M | ESM2-650M | 0.044 (0.007) | 0.024 (0.004) | 0.136 (0.018) | 0.558 (0.009) | 0.036 (0.005) |
| GCN | ESM2-650M | ESM2-650M | 0.052 (0.007) | 0.024 (0.003) | 0.177 (0.020) | 0.576 (0.010) | 0.039 (0.004) |
| GraphSAGE | ESM2-650M | ESM2-650M | 0.055 (0.007) | 0.028 (0.003) | 0.164 (0.018) | 0.571 (0.009) | 0.043 (0.005) |

## B.2 Link prediction baseline

While the majority of existing studies focus on node-level prediction, i.e., predicting which residues are likely to be the epitope residues, we are interested in predicting the interactions between epitope and antigen residues. We argue that, on the one hand, this would provide a more comprehensive understanding of the interaction between epitopes and antigens, and on the other hand, it would be good in terms of model interpretability. Existing methods for predicting epitope residues are mostly based on sequence information, which is not directly interpretable in terms of the interaction between epitopes and antigens.

Our hyperparameter search was conducted within a predefined space as defined in Appendix A.6. We used the Bayesian optimization strategy implemented through Weights & Biases, targeting the maximization of the average bipartite graph link Matthew's Correlation Coefficient (MCC).

The optimization process was managed using the early termination functionality provided by the Weights & Biases' Hyperband method (Falkner et al., 2018), with a range of minimum to maximum iterations set from 3 to 27.

The best performance on both tasks and dataset splits was achieved using a combination of 2 `GCNConv`-layer architecture with `AntiBERTy` and `ESM2-650M` as the antibody and antigen node embeddings, respectively.

For the epitope prediction task, the best-performing model on the epitope-to-antigen surface ratio dataset split has model parameters that include a weight for positive edges of $140.43$, a weight for the sum of positive links of approximately $1.03 \times 10^{-5}$, and an edge cutoff of $3.39$ residues for epitope identification. The decoder applied is an inner-product operator. For the epitope group dataset split, the best-performing model uses an inner-product decoder with a weight for positive edges set to $122.64$, a weight for the sum of positive links of approximately $1.97 \times 10^{-7}$, and an edge cutoff of $9.11$ residues for epitope identification. See Table S7 for their performance.

For the bipartite link prediction task, the best-performing model on the epitope-to-antigen surface ratio dataset split shares the same configuration as the best-performing model for the epitope prediction task on this split. On the epitope group dataset split, the best-performing model uses a weight for positive edges of $125.79$, a weight for the sum of positive links of approximately $3.99e - 7$, and an edge cutoff of $18.56$ residues, also employing an inner-product decoder. See Table S7 for their performance.

# C  Appendix: Ablation Studies

To investigate the impact of different components on WALLE's performance, we carried out ablation studies and described them in this section. For each model variant, we performed hyperparameter tuning and reported the evaluation performance using the model with the best performance on the validation set.

## C.1  Ablation study: replace graph component with linear layers

To investigate whether the graph component within the WALLE framework is essential for its predictive performance, we conducted an ablation study in which the graph component was replaced with two linear layers. We refer to the model as 'WALLE-L'. The first linear layer was followed by a ReLu activation function. Logits output by the second linear layer were used as input to the decoder. The rest of the model architecture remained the same.

It differs from the original WALLE model in that the input to the first linear layer is simply the concatenation of the embeddings of the antibody and antigen nodes, and the linear layers do not consider the graph structure, i.e., the spatial arrangement of either antibody or antigen residues. The model was trained using the same hyperparameters as the original WALLE model. The performance of WALLE-L was evaluated on the test set using the same metrics as the original WALLE model.

## C.2  Ablation study: WALLE with simple node encoding

The presented WALLE model utilizes embeddings from large language models, including ESM2 (Lin et al., 2022) or IgFold(Ruffolo et al., 2023) for representing amino acid sequences, as these models are able to capture the sequential and structural information inherent in protein sequences, providing a rich, context-aware representation of amino acids. To test the effectiveness of such embeddings in this downstream task, we conducted an ablation study where we replaced the embeddings from language models with simple node encodings. Specifically, we evaluated the performance of WALLE when using 'one-hot' encoding and 'BLOSUM62' encoding for amino acids in both antibody and antigen sequences.

## C.3  Ablation study: WALLE with ESM2 embeddings for both antibodies and antigens

We also investigated whether the choice of language models can impact the predictive performance of WALLE; we conducted an ablation study to evaluate the performance of WALLE when both antibodies and antigens are represented using embeddings from the ESM2 language model (Lin et al., 2022) while the original model uses AntiBERTy (Ruffolo et al., 2023) for antibodies as it is trained exclusively on antibody sequences. This also tests whether a language model trained on general protein sequences can be used for a downstream task like antibody-antigen interaction prediction.

**One-hot encoding**

One-hot encoding is a method where each residue is represented as a binary vector. Each position in the vector corresponds to a possible residue type, and the position corresponding to the residue present is marked with a 1, while all other positions are set to 0. This encoding scheme is straightforward and does not incorporate any information about the physical or chemical properties of the residues. This method tests the model's capability to leverage structural and relational information from the graph component without any assumptions introduced by more complex encoding schemes.

**BLOSUM62 encoding**

BLOSUM62 (Henikoff & Henikoff, 1992) encoding involves using the BLOSUM62 matrix, which is a substitution matrix used for sequence alignment of proteins. In this encoding, each residue is represented by its corresponding row in the BLOSUM62 matrix. This method provides a more nuanced representation of residues, reflecting evolutionary relationships and substitution frequencies.

## C.4 Hyperparameter tuning

We used the same hyperparameter search space defined in Appendix A.6 and performed a hyperparameter search as defined in Appendix B.2 for each model variant in the ablation studies. We report the evaluation performance of the tuned model for each variant in Table S9.

Table S9: Performance of WALLE without graph component and simple node encodings on test set from dataset split by epitope to antigen surface ratio.

| Method | Encoding | MCC | AUCROC | Precision | Recall | F1 |
|---|---|---|---|---|---|---|
| WALLE | Both | **0.264** (0.021) | **0.680** (0.014) | **0.258** (0.0117) | 0.534 (0.027) | **0.322** (0.019) |
| WALLE-L | Both | 0.159 (0.016) | 0.612 (0.011) | 0.175 (0.011) | 0.470 (0.024) | 0.237 (0.014) |
| WALLE | ESM2 | 0.196 (0.021) | 0.622 (0.014) | 0.228 (0.019) | 0.410 (0.029) | 0.255 (0.019) |
| WALLE-L | ESM2 | 0.145 (0.014) | 0.610 (0.010) | 0.160 (0.010) | 0.536 (0.022) | 0.227 (0.013) |
| WALLE | One-hot | 0.097 (0.009) | 0.583 (0.008) | 0.119 (0.005) | **0.892** (0.012) | 0.203 (0.008) |
| WALLE | BLOSUM | 0.085 (0.010) | 0.574 (0.008) | 0.118 (0.006) | 0.840 (0.015) | 0.199 (0.008) |

The values in parentheses represent the standard error of the mean; 'WALLE-L' refers to WALLE with the graph component replaced by two linear layers. 'ESM2' refers to the embeddings from the ESM2 language model `esm2_t12_35M_UR50D`. 'One-Hot' refers to one-hot encoding of amino acids. 'BLOSUM62' refers to the BLOSUM62 encoding of amino acids. 'Both' refers to embedding antibodies and antigens using the `esm2_t12_35M_UR50D` ESM2 model and AntiBERTy (via IgFold) language model, respectively. The best performing model is highlighted in bold.

We observed that WALLE's performance with simple node encodings ('one-hot' and 'BLOSUM62') is considerably lower than when using advanced embeddings from language models. This indicates that the embeddings derived from language models capture more nuanced information about the amino acids, enabling the model to better predict epitope-antigen interactions.

The degenerated performance of WALLE with simple encodings can be attributed to the lack of contextual information and structural features in these representations. The high recall but low precision values suggest that the model is unable to distinguish between true and false interactions, leading to a high number of false positives. This highlights the importance of using meaningful embeddings that capture the rich structural and sequential information present in protein sequences.

When comparing WALLE with WALLE-L (without the graph components), we observe that the model's performance drops considerably when the graph component is replaced with fully connected linear layers. This indicates that the topological information captured by the graph component also contributes to the model's predictive performance.

We also observed that WALLE with ESM2 embeddings for both antibodies and antigens achieved similar performance to WALLE with AntiBERTy and ESM2 embeddings for antibodies and antigens, respectively. This suggests that the ESM2 embeddings somehow provide effective information for both antibodies and antigens without training exclusively on antibody sequences.

## D  Impact Statement

This work extends to the optimization of antibody drug development, offering a more efficient and accurate method for predicting where antibodies bind on antigens, a crucial challenge of developing therapeutic antibodies. The potential impact of this advancement is underscored by the recent approval of 136 antibody-based drugs in the United States or European Union, with 17 novel antibody therapeutics approved since January 2023 and 18 more currently under review (Antibody Society, Antibody therapeutics approved or in regulatory review in the EU or US, `https://www.antibodysociety.org/resources/approved-antibodies/`, 24 January 2024). These figures show the transformative impact that antibody design innovation can have on transforming the landscape of therapeutic development, offering new avenues for targeted, effective treatments that can be developed in a more time- and cost-effective manner. Such advancements hold immense potential in accelerating the creation of targeted therapies and implicitly support the broader goals of personalized medicine. Fast-tracking antibody design through *in silico* epitope prediction can enable quicker responses to emerging health threats like pandemics or rapidly mutating pathogens. However, in silico predictions should not be blindly trusted and should merely guide and streamline research and diligent testing, not replace it.

This paper presents work whose goal is to highlight antibody-specific epitope prediction as a major challenge that has not been widely studied to date, which is presented as a bipartite graph connectivity prediction task. This research stands out primarily for its development of a novel benchmarking dataset for antibody-specific epitope prediction - a resource previously unavailable in the scientific community with the potential to set a new standard in the field.

Existing methods are compared using uniform metrics and a vast room for improvement can be demonstrated across the board. A novel model is presented, which surpasses its counterparts by an order of magnitude, outperforming them by at least 5 times, validating its graph-based approach utilizing antibody and antigen structural information as well as leveraging protein language models. We provide a foundational framework that invites other researchers to further build upon and refine this baseline model, which we have demonstrated to be a highly effective approach for this task.

Open availability of the dataset and model facilitates further research and exploration in this domain, expediting the development of more advanced models. Highlighting types of antibody-antigen interactions that are disregarded in existing datasets and methods encourages the examination of shortcoming of current models. The focus on conventional antibodies in this work lays the groundwork for future exploration into epitope prediction for novel antibodies, such as nanobodies, expanding upon potential applications.

## Supplementary Materials

### Dataset Documentation and Intended Uses

We provide a data card for this dataset, `DataCard-AsEP.md`, which can be downloaded using this link: `https://drive.google.com/file/d/1fc5kFcmUdKhyt3WmS3OoLLPgnkyEeUjJ/view?usp=drive_link`

This dataset provides a unified benchmark for researchers to develop new machine-learning-based methods for the epitope prediction task.

### Access to the Dataset

There are two alternative sources where users can download the dataset:

- The dataset can be downloaded using the Python interface provided by our GitHub Repository AsEP-dataset. Detailed instructions on how to download the dataset are provided in the README file. Briefly, after installing the provided Python module, `asep`, the dataset can be downloaded by running the following command in the terminal:

  ```
  download-asep /path/to/directory AsEP
  # For example, to download the dataset to the current directory, run
  # download-asep . AsEP
  ```

- The dataset and benchmark are provided through Zenodo at `https://doi.org/10.5281/zenodo.11495514`.
- Code and Dataset interface is provided in our GitHub Repository **AsEP-dataset** at `https://github.com/biochunan/AsEP-dataset`.

### Author Statement

The authors affirm that they bear all responsibility in case of violation of rights, etc., and confirm the data license. The dataset is licensed under the **CC BY 4.0** License (`https://creativecommons.org/licenses/by/4.0/`), which is provided through the Zenodo repository. The code is licensed under the **MIT License** (`https://opensource.org/licenses/MIT`).

### Hosting, Licensing, and Maintenance Plan

The dataset is hosted on Zenodo, which provides a DOI (10.5281/zenodo.11495514) for the dataset. It also comes with a Python interface provided in our GitHub Repository, AsEP-dataset at `https://github.com/biochunan/AsEP-dataset`, where users can submit issues and ask questions. Future releases and updates will be made available through the same channels. As discussed in the main text, the future plan includes expanding the dataset to include novel types of antibodies, such as single-domain antibodies, and providing more sophisticated features for graph representations. The dataset will be maintained by the authors and will be available for a long time.

### Links to Access the Dataset and Its Metadata

The dataset, benchmark, and metadata are provided through Zenodo.

### The Dataset

The dataset is constructed using `pytorch-geometric Dataset` module. The dataset can be loaded using the following code:

```python
from asep.data.asepv1_dataset import AsEPv1Evaluator

evaluator = AsEPv1Evaluator()

# example
torch.manual_seed(0)
```

```
y_pred = torch.rand(1000)
y_true = torch.randint(0, 2, (1000,))

input_dict = {'y_pred': y_pred, 'y_true': y_true}
result_dict = evaluator.eval(input_dict)
print(result_dict)  # got {'auc-prc': tensor(0.5565)}
% \end{lstlisting}
```

We also provide detailed documentation of the dataset content on Zenodo and include a description below:

- `asepv1-AbDb-IDs.txt`: A text file containing the AbDb identifiers of the 1723 antibody-antigen pairs in the dataset.

- `asepv1_interim_graphs.tar.gz`: Contains 1723 .pt files, where each file is a dictionary with structured data:

  **abdbid** A string representing the antibody AbDb identifier.

  **seqres** A dictionary containing:

  - **ab** An OrderedDict mapping string chain labels `H` and `L` to their corresponding sequence strings, representing heavy and light chains respectively.
  - **ag** A dictionary mapping string chain labels to their corresponding sequence strings.

  **mapping** Includes:

  - **ab** Contains:
    - **seqres2cdr** A binary numpy array indicating the CDR positions in the antibody sequence.
  - **ag** Contains:
    - **seqres2surf** A binary numpy array indicating the surface residues in the antigen sequence.
    - **seqres2epitope** A binary numpy array indicating the epitope residues in the antigen sequence.

  **embedding** Comprises:

  - **ab** Includes embeddings computed using the AntiBERTy model and ESM2 model for the antibody sequences.
  - **ag** Includes embeddings for the antigen sequences computed using the ESM2 model.

  **edges** Describes the interactions:

  - **ab** A sparse coo tensor representing the binary edges between the CDR residues.
  - **ag** A sparse coo tensor representing the binary edges between the surface residues.

  **stats** Metadata about each antibody-antigen pair, including counts of CDR, surface, and epitope residues, and the epitope-to-surface ratio.

- `structures.tar.gz`: Contains 1723 pdb structures, each named using the AbDb identifier.

- `split_dict.pt`: Contains the `train/val/test` splits of the dataset, with splits based on the epitope ratio and epitope group of the antigen.

**Long-term Preservation**

The current version of the dataset and benchmark are provided through Zenodo, which provides long-term storage. Future versions will be made available through the same channel and users are encouraged to submit queries and issues through the issues channel on the GitHub repository.

**Explicit License**

The dataset is licensed under the **CC BY 4.0** license (`https://creativecommons.org/licenses/by/4.0/`), which is provided through the Zenodo repository. The code is licensed under the MIT License `https://opensource.org/licenses/MIT`.

**Benchmarks**

Detailed benchmark experiments and results are provided on Zenodo (`https://doi.org/10.5281/zenodo.11495514`), and the file `benchmark.zip` contains the instructions on how to reproduce the results. To run ESMFold, EpiPred, and MaSIF-Site, we provided docker images on the Zenodo repository or instructions on how to obtain them from DockerHub. The benchmark results are reproducible by following the instructions provided in the zip file. For the method WALLE and its variants used in the ablation studies, their configuration YAML files and the trained models are also provided on Zenodo.

