# OpenReview forum: "AsEP: Benchmarking Deep Learning Methods for Antibody-specific Epitope Prediction"
_NeurIPS.cc/2024/Datasets_and_Benchmarks_Track — NeurIPS 2024 Track Datasets and Benchmarks Poster_

### Official Review · Reviewer_cuAf · 2024-07-24
**A Solid Framework For Antibody Antigen Epitope Prediction**

**Rating:** 8
**Confidence:** 3

**Review:**

Overall, the paper provides a compelling formulation of the AB/AG epitope prediction task and uses sound biological intuition to construct the dataset. The notation is precise and aids readers' comprehension rather than bogging them down. When introducing a model, it leverages the unique information gain posed by the dataset to create what will be a decent baseline for other works that build on this benchmarking suite. From the scale of the data gathered and improved scope, researchers can easily build upon the dataset and derive worthwhile insights.

### Pros
* This work puts forward a general framing epitope prediction consider both single residue antigen epitope prediction and antibody-antigen epitope edge prediction.
* The introduced dataset is the largest put forth for the task and comes with pre-computed features making it easy to use.
* WALLE, the new model, serves as a quality baseline on this task and improves upon older methods. Its modeling choices are well supported by quality ablation studies.

### Cons
* The dataset only considers antibodies with VH/VL domains while other prominent therapeutic form factors exist (e.g. VHHs/nanobodies).
* Though ablating a fair portion of the modeling choices, certain comparisons are still required for both WALLE and baselines (larger ESM2 and AF2-Multimer).

**Strengths:**

The work at hand does a great job in introducing the problem of antibody antigen epitope prediction and crystallizes a mathematical description of the task and uses norms inline with the biological community to derive residue labels. Readers can easily recreate the generated dataset and the appendix provides clear instructions for the curation. Furthermore, looking at the GitHub repo one can simple code and well-documented instructions for building off of the work. What's introduced is both sound in construction and empathetic in downstream use.

Furthermore, the studies did a great job in constructing a solid baseline for the community and further demonstrating why the bipartite graph structuring of the problem is beneficial in their WALLE ablations. Modeling choices are well motivated by the experiments and hold up as both simple baselines and strong starting points for the community.

**Additional Feedback:**

The work is strong in what it presents and could be strengthened with some minor improvements that I listed above: clarity, an additional baseline, and a larger ESM2 study for WALLE. If the above points get addressed in the work, I'd be happy to raise my score.

**Clarity:**

Aside from the point above about the clarity towards which subtask is being investigated, the paper is quite easy to follow. The narrative builds logically and the collection scheme is clear. It would aid readers if the way that baselines are run got more attention, but the rest is easy to follow. Notably, the math notation introduced does a great deal of good in conveying the work's framing of the epitope prediction problem.

**Correctness:**

All claims made in the paper are supported by the experiments performed. The work does a good job in running ablation studies to motivate the architecture choices in a sound manner. The dataset collection scheme is sensible using a common source of ground truth structures and common biological tools to construct the dataset in a manner which mirrors previous work.

**Documentation:**

As it stands, one could readily recreate the exact pipeline to collect the dataset in the paper at hand. The appendix provides helpful visuals and exact details for how to curate it. Moreover, the work continues to emphasize good user experience through its codebase. The GitHub repository demonstrates clear means to use the dataset and baseline model. It is clear the intended uses of the dataset, but there's no explicit plan for hosting or maintenance.

**Ethics:**

There are no ethical concerns with the work.

**Limitations:**

The authors do an okay job in addressing the limitations of the method and baseline. Some mentioning of where the proposed dataset and models can be improved upon exists, but the treatment is not that thorough. The impact statement does successfully address any societal concerns, and in general this work will hopefully lead to real world improved health outcomes.

**Opportunities For Improvement:**

In general my room for growth falls into three camps: dataset composition, baselines, and clarity.

### Dataset Composition
VHHs for example form a popular therapeutic modality for modern antibody campaigns. It would make this work stronger to include them in the prediction task, but as a reviewer I understand the challenge this poses as it requests you to recreate the entire dataset. One detail that would help is understanding the composition of antibody formats included: IGG, SCFV, bispecifics, or any other noteworthy forms in the form of a pie chart or bar graph in the appendix.

### Baselines
WALLE is constructed quite well, but I do find the usage of ESM2-35M likely to leave performance on the table. Seeing the results using the 650M model, as that's most adopted by the community, would make the WALLE baselines stronger.

Though ESMFold is indeed fast, it's not trained directly for multimer structure prediction. The baseline need AF2-Multimer results to be seen as complete. Given the size of the test set, this ask seems nontrivial, but definitely doable.

### Clarity
My note on clarity is minor as on a whole the text reads very well. The detail that left me scratching my head was which classification subtask was being measured. Since two distinct subtasks were defined, it left me confused as to which subtask each table referred to. It appears that Table 1 is referring to epitope classification at the node level, but I had to cross reference the appendix to be certain. Stating this more apparently in the main text would save readers from juggling the pages of the manuscript.

**Relation To Prior Work:**

The relationship to prior epitope work is made clear and solid initial baselines have been chosen. Past datasets are mentioned, albeit quite briefly in the appendix for those in the table, so it would benefit the work to cover their individual makeup and differences a tad more.

**Summary And Contributions:**

This work puts forth both a significantly larger dataset to perform antibody antigen epitope prediction and a performant GCN model geared towards the task. It extends on existing structures of AB/AG complexes to derive a dataset 3 times larger than past work and also more specific in problem framing. It chooses to model the graphs of the antibody and antigen, denoted $G_A$ and $G_B$ respectively, to study the classification of both the classical residues in the antigen $f:V_B\\to \\{ 0,1 \\}$ as well as the bipartite edges spanning from the antigen to the antibody $g:V_B\\times V_A\\to\\{0,1\\}$. Such a formulation further increases the granularity of the task. Lastly, their proposed model WALLE leverages their created dataset and benchmark to derive a state-of-the-art solution to epitope prediction on this dataset.

---

> ### Author Rebuttal · Authors · 2024-08-16
>
> Thank you for your thoughtful feedback. We acknowledge the significance of expanding our dataset to include other prominent therapeutic forms such as VHHs/nanobodies. As mentioned in the supplementary information, this expansion is part of our future plans and will be addressed in the next dataset release.
>
> Regarding the baselines for WALLE, we appreciate the reviewer's suggestions and agree that incorporating additional comparisons would strengthen the work. Initially, we selected the smaller ESM2-35M model due to its reduced number of parameters and its output size being comparable to AntiBERTy, making it a suitable choice for our baseline. However, to enhance the robustness of our results, we are working on adding the ESM2-650M model as an embedding method and will post the results as soon as we complete it, and it will also be included in the camera-ready version.
>
> We understand the importance of including an AF2-Multimer (AF2M) baseline as well. We have begun running AF2M on the test sets, which comprise 325 unique items from two splits. However, the multiple sequence alignment (MSA) generation step is time-intensive, taking between 2 and 7 hours per item, excluding the structure prediction time. Given these constraints, completing this baseline will likely take approximately 1.5 months. While we are committed to including the AF2M results in the camera-ready version, we may not be able to finalize them before the end of the author/review discussion phase. We appreciate the reviewer's understanding in this matter.
>
> Additionally, we will improve the clarity of our manuscript by adding a short description of the makeup and differences of past datasets in the “Appendix A.1 Related Work” section, as follows:
>
> - Gao et al. 2022: They used the dataset from Zhao et al. (2018) [https://doi.org/10.1093/bioinformatics/bty051] and included 257 antibody-antigen complexes, encompassing both VHVL and VHH antibodies. However, the primary focus of this research is on epitope prediction using the antigen as the input. Consequently, antibodies were not included in the predictive models, rendering the dataset unsuitable for antibody-specific epitope prediction.
> - CSM-AB: As described by their supplementary information, their dataset contains 472 antibody-antigen structures including 375 Fab, 82 Nanobody and 12 scFv. These structures were collected from PDB, identified using Chothia annotation as in Dunbar and Deane, 2016.  The authors did not describe the procedure of any further filtering steps. We assumed they included all available structures at that time with available protein-protein binding affinity information from PDBbind since the study aims to predict binding affinity.
> - SAGERank: The authors of SAGERank formed a dataset composed of 287 antibody-antigen complexes filtered by sequence identity at 95%. While the authors did not explicitly mention the antibody types, we infer from the results that these are also Fab antibodies that include both VH and VL domains. The dataset was composed mainly for docking pose ranking output by MegaDock and did not include interface clustering, i.e. epitope grouping, as we did.
>
> We will also clarify the classification subtasks in the main text to ensure that readers can easily identify which subtask each table refers to without needing to cross-reference the appendix.
>
> Lastly, we acknowledge the reviewer's note on the hosting and maintenance of the dataset. The plan for this is provided in the Data Card in the supplementary information. Both current and future versions of the dataset are and will be hosted on Zenodo, with annual updates to include new antibody-antigen complexes and additional types of antibodies.

---

> > ### Comment · Reviewer_cuAf · 2024-08-23
> > **Rebuttal Response**
> >
> > Each of the points has been thoughtfully addressed and is set to appear in the final draft of the work. WALLE is being benchmarked at an increased ESM-2 size. AlphaFold 2 Multimer is currently in progress and by no means a simple effort, so one must appreciate the care taken in this work to ensure this work has a thorough set of results. Lastly, the supplemental work does a fantastic job with documentation and upkeep.
> >
> > Under the assumption that the in progress experiments are added to the camera ready version, I am further increasing my score as a result of these improvements.

---

### Official Review · Reviewer_TnvS · 2024-07-25
**Review of the submission of the paper titled "AsEP: Benchmarking Deep Learning Methods for Antibody-specific Epitope Prediction"**

**Rating:** 7
**Confidence:** 3
**Clarity:** The paper is well written with proper…

**Review:**

Strength of this work:
1. Introduced a robust dataset.
2. Clearly discussed previous and existing works.
3. Discussed two different ways of dataset split.
4. Discussed the reasoning behind the choice of graph convolutional layers of fully connected layers.
5. Experimentally discussed the reasoning behind using different methods for node embedding generation of antibody and antigen graphs respectively.

Weaknesses:
1. Needs a more detailed description of the deep learning model introduced by the authors. Especially the decoder. Maybe a separate visualization of the inner workings of the decoder would provide more clarity.
2. Node embeddings for the antigen graphs were generated by ESM-2 (esm2_t12_35M_UR50D). However, it is a smaller model, with a much larger version of it with 15 billion parameters available (esm2_t48_15B_UR50D). Is there any particular reason why this model was not considered for generating node embeddings?
3. Authors should provide more information regarding the choice of GCN. What was the motivation behind using GCN compared to other graph neural network modules such as GraphSAGE?

**Strengths:**

The primary strength of this paper is the introduction of a benchmark dataset for the task of antibody-specific epitope prediction. The authors clearly describe in detail how the data was curated and organized. The authors also discuss the two different dataset splits, which provides robustness to their analysis. The authors also clearly highlight existing work, and how the authors' work builds upon them. The authors highlighted the reasoning behind using graph convolutional layers over fully connected layers. The authors also experimented with different embedding generation methods for both the antibody and the antigen graph and were transparent about their choice of using AntiBERTy and ESM-2 for antibody and antigen graphs respectively.

**Additional Feedback:**

No additional feedback is needed.

**Correctness:**

According to the details provided in this paper, the dataset seems to be constructed soundly. The authors also highlight two different ways of dataset split which provides clarity and robustness in their dataset curation process. The authors also provide detailed evaluations based on different metrics of their proposed model and other existing models.

**Documentation:**

The authors provide sufficient details on how to access the dataset and also provide details on how the entire data curation process was implemented.

**Ethics:**

To the best of my knowledge, the submission raises no ethical concerns.

**Limitations:**

The authors are very transparent about the limitations and the scope of future improvements for this particular work. The authors highlight the need for a more sophisticated model to get further improvements. The authors are very transparent about their proposed method being biased towards one dataset split over the other, and the authors stated that the possible reasoning behind it could be inherent biases of their proposed deep learning method towards the training dataset. The authors also highlight the possibility of future work incorporating non-conventional antibodies such as nanobodies in their dataset curation.

**Opportunities For Improvement:**

The authors should provide more details about the decoder of the model section as the existing description seems ambiguous. Is it merely a function that provides output based on probability thresholds or is it a trainable weighted operator like conventional neural network layers?

The authors do highlight the need for GCN over traditional fully connected layers for the processing of embeddings. However, the authors could have experimented more in this regard with other graph neural network operators such as GraphSAGE, GATConv, etc. The authors mention PyTorch Geometric as the primary Python package used for this work. This package contains the modules for the graph neural network operators such as GraphSAGE, GATConv, etc.

For the node embeddings of the antigen graph, the authors utilized a smaller version of the ESM-2 model. There is a larger version of this model available (esm2_t48_15B_UR50D). The authors should provide the reasoning behind opting for a smaller model.

**Relation To Prior Work:**

The authors discuss extensively how their work differs from previous works in this field and also the authors highlight their new contribution in this field.

**Summary And Contributions:**

The paper is an interesting work drawing attention to the field of antibody-specific epitope prediction, where the authors curate a benchmarking dataset as well as apply a graph neural network-based model for the dual task of epitope prediction and bipartite link prediction between antibody and antigens. The primary contribution of this work is the benchmarking dataset curated by the authors as well as the deep learning method WALLE which is a graph neural network-based method.

---

> ### Author Rebuttal · Authors · 2024-08-16
>
> Thank you for your thoughtful feedback. We agree that further clarification of the decoder is necessary to enhance clarity, and we will address this in the camera-ready version (specifically in subsection 'Decoder' under Section 5). To elaborate, WALLE employs a straightforward decoder that calculates the inner product of the antibody and antigen node embeddings to generate the output. Additionally, during hyperparameter tuning, we experimented with a more complex decoder structure, incorporating a fully connected layer with a bias term and a dropout rate of 0.1. This alternative decoder yielded comparable performance to the simpler inner product approach (as detailed in Appendix A.6).
>
> The intention behind the WALLE model is to establish an efficient and viable baseline model. However, we recognize the value of the reviewer's suggestions and will incorporate them in the camera-ready version. Specifically, we will include the ESM2-650M model for generating node embeddings and the GraphSAGE architecture as a comparison to our current approach. We are currently working on this and will post the results as soon as we complete them and these will be included in the camera-ready version.
>
> While we acknowledge that larger language model embeddings may offer additional benefits, we also recognize some recent findings. In particular, recent work by Kevin K. Yang presented at ICML2024, "Feature Reuse and Scaling: Understanding Transfer Learning with Protein Language Models" (https://proceedings.mlr.press/v235/li24a.html), suggests that scaling PLM pretraining does not necessarily improve performance on downstream tasks. However, we are committed to exploring these avenues further in our upcoming work. Thank you again for your valuable feedback, which will help us to refine and strengthen our work.

---

> > ### Comment · Reviewer_TnvS · 2024-09-06
> >
> > Thank you for the response. The authors highlighted how they will incorporate the feedback into the camera-ready version and also answered any concerns properly.

---

### Official Review · Reviewer_JQrQ · 2024-07-30
**A well constructed benchmark for epitope prediction**

**Rating:** 7
**Confidence:** 4
**Clarity:** Yes, the dataset is well presented an…

**Review:**

The overall dataset is well-constructed, with a clear presentation of the approach and major details. This dataset bridges existing research gaps and will benefit the broader AI for drug discovery community. The contributions include the construction of the dataset, framing the epitope prediction task as a bipartite link matching problem, benchmarking on several existing model classes, and proposing a new architecture that combines the strengths of some of these models. Although other antibody-antigen datasets exist, there are no other datasets that is tailored for the epitope prediction task. Overall, the contribution is significant, and the benchmark has the potential to create a meaningful impact.

**Strengths:**

- **Well-Constructed Dataset:** The dataset is well constructed and clearly presented.
- **Addresses Research Gaps:** It bridges existing research gaps in epitope prediction. The dataset will benefit the broader AI for drug discovery community.
- **Comprehensive Contributions:**
  - Construction of a new dataset for epitope prediction.
  - Framing the epitope prediction task as a bipartite link matching problem.
  - Benchmarking on several existing model classes.
  - Introducing a novel architecture that combines the strengths of these models.

**Additional Feedback:**

None.

**Correctness:**

The dataset is constructed soundly. Since the structures are sourced from PDB, there are no potential issues with correctness, and several samples have been filtered out to ensure reliability. The evaluation design and methods are robust and well-executed.

**Documentation:**

The dataset is well documented, and there is sufficient detail to support reproducibility. The authors have made the dataset in a publicly accessible forum.

**Ethics:**

There are no ethical concerns with the dataset.

**Limitations:**

The authors have discussed limitations throughout the paper, but there is no dedicated section for them. I suggest that the authors consider including additional limitations:

1. **Dataset Domain Limitations:** Does the dataset adequately cover the antibody design space? Are there any anticipated challenges in training ML models, such as issues with generalization or the risk of learning spurious correlations?
2. **Problem Formulation:** Does reducing the problem to a bipartite matching framework result in any loss of information that could complicate model training?

**Opportunities For Improvement:**

- The authors should clearly articulate the section on the limitations (see the limitations section).
- Expand the related work section to include other antibody-antigen datasets used in building ML models.
- Include pointers for future work to build better models.

**Relation To Prior Work:**

Yes, the overall novelty of the proposed benchmark is well-stated. However, the authors should consider expanding their related work to include other antibody structure datasets, particularly those used in ML binding prediction and generative model tasks.

**Summary And Contributions:**

The paper constructs a new dataset for the task of epitope prediction. While the task of protein/antibody generation is gaining increasing attention in recent research, the fundamental problem of epitope prediction is understudied, and this benchmark aims to address that gap. The dataset is constructed from a subset of the Antibody Dataset (AbDb), after filtering out unresolved samples, and several dataset splits are presented. The epitope prediction task is framed as a bipartite matching problem. Additionally, the proposed dataset has been benchmarked on several model classes, and a simple novel architecture that is meaningul for the proposed setup is introduced.

---

> ### Author Rebuttal · Authors · 2024-08-16
>
> Thank you for your valuable feedback. We agree that a dedicated section on limitations would enhance the clarity and completeness of our work. We will add a “Limitations” section in the camera-ready version of the manuscript, addressing both the points raised by the reviewer and other relevant aspects.
>
>
> **Generalization and Dataset Domain Limitations**: We acknowledge the inherent challenges in ensuring that machine learning models generalize well to new data, especially in the complex domain of antibody design. To mitigate the risk of poor generalization, we have implemented two key strategies:
>
> 1. **Epitope-Based Dataset Splitting**: We employed a dataset split strategy based on `epitope groups,’ which allows us to test methods on unseen epitopes. This approach simulates real-world scenarios where novel antibody-antigen pairs and previously unrepresented epitopes are encountered, thereby providing a more robust assessment of each method’s ability to generalize.
> 2. **Expanding Dataset Coverage**: Our dataset is currently the largest available, covering a broad spectrum of the antibody design space. However, we recognize that this space is continually evolving. As such, we have committed to updating the dataset annually with new structures as they become available, and expanding it to include novel antibody types such as nanobodies, heavy-chain- or light-chain-dimer antibodies. These updates, detailed in the data card within the supplementary materials, will further enhance the dataset’s coverage and utility.
>
> **Problem Formulation and Information Retention**: Regarding the use of the bipartite matching framework, we want to clarify that this framework is employed solely as an evaluation step and does not result in any loss of information during model training. The model retains full access to the detailed individual graphs of both antibodies and antigens, including all relevant node features and edges. This comprehensive access ensures that all critical information is preserved and utilized during training, thereby maintaining the integrity and effectiveness of the model. Additionally, when users download our dataset, it comes with a tar file that includes the structure files and raw graphs, enabling users to customize their embedding approach according to their specific needs.
>
> In addition to the above, we will also expand the related work section to include other antibody-antigen datasets that have been used in building machine learning models. Furthermore, we will provide pointers for future work, highlighting potential avenues for building better models and further improving the predictive performance and generalizability of the current approach.
>
> Thank you again for your constructive suggestions, which will help us refine and strengthen our manuscript.

---

> > ### Comment · Reviewer_JQrQ · 2024-08-29
> > **Thank you for the response**
> >
> > Thank you for the response. I request the authors to incorporate the feedback in the camera ready version.

---

### Official Review · Reviewer_bJmy · 2024-07-30
**Review: AsEP**

**Rating:** 7
**Confidence:** 4
**Correctness:** Yes, both the dataset and the evaluat…
**Clarity:** Yes, the paper is easy to follow and …

**Review:**

The paper is easy to read and follow. The dataset splits and training process are clearly detailed.

**Strengths:**

While it is already known that incorporating sequence and structure features provide insight into antibody–antigen recognition problems, this work is relevant to the research community as:
1. This work presents a large dataset (1723 AbAg complexes) than previous released dataset.
2. The dataset is packaged as PyTorch Geometric dataset for easy loading and training models.
3. The authors showed better results on MCC metric when compared with other methods.

**Additional Feedback:**

Overall, I find this paper interesting and impactful. The dataset is available as dataset object and the authors have propose a graph neural network based method- WALLE. An explanation on how does WALLE differ from previous methods would make the paper better.

**Documentation:**

The GitHub repository provides the details to support reproducibility. The dataset is hosted in Zenodo with a DOI.

**Ethics:**

No ethics concerns.

**Limitations:**

The authors have discussed the limitations and societal impacts, which is convincing for me.

**Opportunities For Improvement:**

Benchmarking more models strengths the paper.

**Relation To Prior Work:**

Yes, the paper discusses how the dataset differs from previously released dataset.

**Summary And Contributions:**

This paper presents a filtered anitbody-antigen complex structure dataset aimed for developing and testing new epitope prediction methods. Users can load the dataset as a PyTorch Geometric dataset object including an option to load the node embeddings derived from AntiBERTy and ESM2. The dataset contains both node and edge labels. The authors then trained a graph-based model called WALLE that contains two separate GCN modules for the antibody and antigen to capture it's their structural features which are then passed into decoder module for final prediction. This paper provides contribution by putting forward a large dataset in comparison to previous datasets and the dataset is used to train and test the GNN module.

---

> ### Author Rebuttal · Authors · 2024-08-16
>
> Thank you for your suggestions. We appreciate your recommendation to clarify how WALLE differs from previous methods. While we described how existing methods approach this problem in the related work section, we realize we did not explicitly highlight the distinctions of our approach.
>
> WALLE combines a graph-based approach with learned features from a protein language model, which sets it apart from existing methods that either rely purely on sequence-based information without incorporating structural data (such as ESMBind and ESMFold) or use structural information in graph-based methods without leveraging the rich embeddings provided by protein language models (such as PECAN, EPMP, EpiPred, or MaSIF-site). By integrating protein-language model embeddings with a structural graph protein representation, WALLE provides a more comprehensive and information-rich approach to epitope prediction. We will clarify this distinction in Section 5, “WALLE: a graph-based method for epitope prediction,” to ensure that readers fully understand the novel aspects of our approach.
>
> Regarding your suggestion on benchmarking more models to strengthen the paper, we are currently working on adding the ESM2-650M model for node embedding and experimenting with the GraphSAGE architecture, as suggested by the other reviewers. Additionally, we are in the process of benchmarking AF2-Multimer (AF2M) as part of our ongoing efforts to provide a more comprehensive evaluation of WALLE against other models. We will post the results as soon as they are completed, and these additions will be included in the camera-ready version.

---

> > ### Comment · Reviewer_bJmy · 2024-08-28
> > **Rebuttal Response**
> >
> > The authors have addressed the feedback and are actively working to enhance the paper by incorporating additional experiments. I acknowledge that running these experiments and benchmarks requires significant time and effort. With the authors’ commitment to including these improvements in the camera-ready version, I am raising my score. This is a good work!

---

### Decision · Program_Chairs · 2024-09-26

**Decision:**

Accept (Poster)

**Comment:**

All the reviewers agree the AsEP benchmark is a significant and valuable contribution to antibody-specific epitope prediction.

The main pros include: (1) the largest antibody epitope prediction dataset; (2) precomputed features easy for users to use; (3) a newly developed prediction model, WALLE, that can serves as a strong baseline for future development; (4) framing the epitope prediction task as a bipartite link matching problem; and (5) benchmarking on several existing model classes and introducing a new architecture to combine the strengths of different models.

The main cons include: (1) lack of benchmarking with AlphaFold-multimer; (2) having not used larger LLMs to generate features, and (3) lack of some other epitope prediction data such as nanobody data. The authors are conducting experiments to address (1) and (2) and are committed to add them into the final version of the paper. They also plan to continue to expand the dataset periodically to address (3).

Overall, the benchmark is an important contribution to the field and can stimulate the development of more antibody epitope prediction methods.